# Advances in Mass Spectrometry-Based Blood Metabolomics Profiling for Non-Cancer Diseases: A Comprehensive Review

**DOI:** 10.3390/metabo14010054

**Published:** 2024-01-14

**Authors:** Ekaterina Demicheva, Vladislav Dordiuk, Fernando Polanco Espino, Konstantin Ushenin, Saied Aboushanab, Vadim Shevyrin, Aleksey Buhler, Elena Mukhlynina, Olga Solovyova, Irina Danilova, Elena Kovaleva

**Affiliations:** 1Institute of Natural Sciences and Mathematics, Ural Federal University, Ekaterinburg 620075, Russia; vladislav0860@gmail.com (V.D.); fpolancoespino@gmail.com (F.P.E.); konstantin.ushenin@urfu.ru (K.U.); zellist@mail.ru (A.B.); elena.mukhlynina@yandex.ru (E.M.); soloveva.olga@urfu.ru (O.S.); ig-danilova@yandex.ru (I.D.); 2Institute of Immunology and Physiology of the Ural Branch of the Russian Academy of Sciences, Ekaterinburg 620049, Russia; 3Autonomous Non-Profit Organization Artificial Intelligence Research Institute (AIRI), Moscow 105064, Russia; 4Institute of Chemical Engineering, Ural Federal University, Ekaterinburg 620002, Russia; sabushanab@urfu.ru (S.A.); vadim.shevyrin@gmail.com (V.S.); e.g.kovaleva@urfu.ru (E.K.)

**Keywords:** blood profiling, mass spectrometry, GC×GC-MS, non-cancer diseases, metabolomics, lipidomics, glycomics, biomarkers, review

## Abstract

Blood metabolomics profiling using mass spectrometry has emerged as a powerful approach for investigating non-cancer diseases and understanding their underlying metabolic alterations. Blood, as a readily accessible physiological fluid, contains a diverse repertoire of metabolites derived from various physiological systems. Mass spectrometry offers a universal and precise analytical platform for the comprehensive analysis of blood metabolites, encompassing proteins, lipids, peptides, glycans, and immunoglobulins. In this comprehensive review, we present an overview of the research landscape in mass spectrometry-based blood metabolomics profiling. While the field of metabolomics research is primarily focused on cancer, this review specifically highlights studies related to non-cancer diseases, aiming to bring attention to valuable research that often remains overshadowed. Employing natural language processing methods, we processed 507 articles to provide insights into the application of metabolomic studies for specific diseases and physiological systems. The review encompasses a wide range of non-cancer diseases, with emphasis on cardiovascular disease, reproductive disease, diabetes, inflammation, and immunodeficiency states. By analyzing blood samples, researchers gain valuable insights into the metabolic perturbations associated with these diseases, potentially leading to the identification of novel biomarkers and the development of personalized therapeutic approaches. Furthermore, we provide a comprehensive overview of various mass spectrometry approaches utilized in blood metabolomics research, including GC-MS, LC-MS, and others discussing their advantages and limitations. To enhance the scope, we propose including recent review articles supporting the applicability of GC×GC-MS for metabolomics-based studies. This addition will contribute to a more exhaustive understanding of the available analytical techniques. The Integration of mass spectrometry-based blood profiling into clinical practice holds promise for improving disease diagnosis, treatment monitoring, and patient outcomes. By unraveling the complex metabolic alterations associated with non-cancer diseases, researchers and healthcare professionals can pave the way for precision medicine and personalized therapeutic interventions. Continuous advancements in mass spectrometry technology and data analysis methods will further enhance the potential of blood metabolomics profiling in non-cancer diseases, facilitating its translation from the laboratory to routine clinical application.

## 1. Introduction

Blood analysis occupies a prominent part within the domains of clinical diagnostics and scientific research, particularly from the perspective of mass spectrometry (MS) [1,2]. Traditionally, the examination of human venous blood involves substantial volumes, often in the range of 100–200 mL, with adjustments made in animal studies according to the species under investigation. Following initial collection, centrifugation separates the whole blood components into plasma and red blood cells, preparing the ground for the subsequent analysis [1]. The next phase entails the utilization of solvents and detergents to extract a variety of bioactive substances, including proteins, glycans, peptides, lipids, and amino acids, from serum or plasma. This carefully extracted array of compounds is subjected to separation by advanced techniques such as chromatography or electrophoresis, although MS stands as the foundation of this investigative process.

As an analytical technique, MS ionizes molecules within the sample, speeds them up, and evaluates the ratio of their mass to charge [3]. This methodology aids not only in the identification of specific molecules within the sample but also in the precise quantification of their abundance through robust statistical data processing. In the context of this review, we undertook a thorough exploration of a rich collection of published data related to blood sample processing via MS. Notably, over 80% of the studies were primarily concentrated on cancer, oncology, and carcinogenesis. While cancer reach is undeniably of essential significance, this exclusive focus has at times overshadowed the other crucial applications of MS in blood analysis.

Numerous methodologies for non-cancer disease profiling have achieved a level of development that enables their direct translation into clinical practice. Therefore, this review is strategically focused on 814 articles that employ MS for blood profiling in both fundamental and applied research, including GC×GC-MS for metabolomics-based studies. To navigate through the hard task of analyzing such a vast number of studies, we utilized natural language processing methods in order to cluster the studies based on their thematic content. Additionally, trainable text classifiers were deployed to enhance our analysis of this multifaceted research domain. Our methodology is described in the Section 2, and the basic scientometric analysis of the field is presented in the Section 3.

Within our manual analysis, we organized our findings into three distinct sections. The Section 4 offers a brief explanation of MS methods to facilitate a better comprehension of the technical aspects. The other two sections are devised to reflect the contents of the studies found. Thus, MS, along with metabolomics applications in blood profiling, could be viewed from two perspectives: as a certain class of metabolite studies or with reference to certain diseases. Therefore, we offer two principal sections of studies according to these two perspectives. The first section, entitled Research Field Landscape, presents an exploration of various fields of metabolomics, based on studied metabolites classes, such as ’lipidomics’, and ’glycomics’. This section provides definitions for each area of metabolomics that is integrated in some way with blood profiling using the MS approach. It explores the structure of analyzed substances, their functional roles in the organism, and their involvement in the pathogenesis of non-cancer diseases; moreover, it outlines the clinical applications. The second section, Disease Study Landscape, contains a comprehensive review of several socially significant non-cancer diseases of inflammatory, bacterial, rheumatoid, and other natures. Each disease is provided with a brief summary, marked symptoms, and known methods of diagnostics. Each disease description includes particular metabolites that are often either not analyzed in the profiling of the diseases reviewed or serve as other potential compounds. All such metabolites are reviewed in the Section 5. Therefore, we fill the gap between the application of known metabolites and their potential usage in the diagnosis and management of non-cancer diseases.

This review is intended to serve as a needed resource for researchers, clinicians, and practitioners to enable them to take advantage of the MS approach in the realm of non-cancer blood metabolomics. It provides critical insights, highlights methodological advancements, reveals promising opportunities beyond the predominant cancer-centric paradigm, and remarks on the incorporation of GC×GC-MS for metabolomics-based studies.

## 2. Review Methodology

In the first step, the scope of the review is outlined as a list of search tags. Most research groups do not explicitly state that they conduct metabolomics studies, but specify their research field as lipidomics and glycomics. Thus, we picked all the keywords and extended the list with synonyms. For example, we would assume that the article was about lipidomics if its abstract included words such as lipidomics, lipidome, or lipid profiling. In the same way, we compiled a list of spectrometry acronyms (GC-MS, LC-MS, etc.) and unfolded each acronym to its full name with various forms. Finally, we chose the diseases of interest: systemic lupus erythematosus, tuberculosis, sepsis, and rheumatoid arthritis. Also, we added general words such as immunodeficiency, immunodeficient state, and inflammation to the search tags. A list of exception tags was drawn up to exclude unnecessary articles. The list included genomics, pharmacology, and food studies because these topics fell outside the review scope. Table 1 shows 35 tags for the literature search.

In the second step, we downloaded article abstracts and meta-information from the PubMed^®^ database of the National Institutes of Health, United States National Library of Medicine, Bethesda, MD, USA. The search was confined to the period from 1 January 2003 to 31 March 2023. Overall, 871 abstracts were obtained at this step. Then, we excluded from consideration preprints, dissertations, posters, presentations, conference papers, and abstracts without full text. Thus, the review involved only full-text articles, based on the assumption that full-length articles would undergo editorial evaluation, rendering information for analysis more trustworthy. This stage provided us with 560 full-text articles. However, thereafter, we decided to focus on a detailed analysis of research works published in the last decade, characterized by a sharp rise in the number of studies, which reduced our pool to 507 articles.

Then, a natural language processing approach was used in order to locate the text groups with semantic similarities [4]. Firstly, standard text preprocessing methods were applied, such as tokenization, lemmatization, and stop word removal. After that, the texts were encoded by means of the term frequency and inverse document frequency approach [5]. For more advanced analysis, we encoded text using BlueBERT [6], a large language model trained on PubMed abstracts and clinical notes. Both approaches allowed us to present texts as vectors in a certain space, where texts with similar meanings are located near each other.

To reduce the number of vector components, various methods were applied, including principal component analysis, t-SNE [7], UMAP [8], and PHATE [9]. The results were visualized as interactive plots using the Plotly Python package. The authors of this review manually explored these interactive visualizations and categorized groups of articles according to their fields of expertise. In total, 16 groups of interest were identified, including oxylipins, phospholipids, glycerophospholipids, sphingolipids, sterol lipids, N-glycans, O-glycans, amino acids, tuberculosis, non-alcoholic fatty liver disease, systemic lupus erythematosus, sepsis, human immunodeficiency virus, rheumatoid arthritis, diabetes, and a small group of various other diseases.

In the last stage of the review process, we added several important textbooks and articles that were necessary to enhance reader comprehension. And we included information from publications available through Open Access, Affiliated Access, Research Gate^®^, San Francisco, CA, USA, and those supplied upon request. The entire text selection pipeline is depicted in Figure 1.

## 3. Scientometrics

We analyzed 560 full-text articles published in the last 20 years. As can be seen from Figure 2, the number of publications in the field of research under consideration has been growing exponentially. Minor declines took place in 2006, 2010, 2012, 2015, and 2018. We suggest this may be related to the deterioration of the macroeconomic environment in 2004, 2008, 2013, and 2017, because research productivity on a global scale depends on the world economy.

Figure 3 and Table 2 show the text numbers for each field of research and each disease from the last 10 and 20 years. As noted in the foregoing, metabolomics includes other research fields but is usually mentioned in texts that focus on specific diseases. As we can see, the most popular field turned out to be metabolomics, with 386 texts published in the last decade. We believe this may be related to the growing interest in the application of mass spectrometry for disease detection and profiling. Next, following metabolomics, we have texts about lipidomics and glycomics with 113 and 8 published studies, respectively. We suggest that these numbers indicate a degree of universality within the research field. Lipidomics is more popular than glycomics because lipid profiles are associated with inflammation, cardiovascular disease, obesity, etc. All of them are widespread and socially important diseases. At the same time, changes in the glycome are observed only in a limited number of health conditions.

## 4. Methods Landscape

### 4.1. Separation Methods

Chromatography stands as a primary method for the separation of compounds in metabolomic analysis. Gas chromatography and liquid chromatography variants are preferred, and are seamlessly integrated with mass spectrometers. In addition to chromatographic methods, capillary electrophoresis has gained prominence in certain instances.

#### 4.1.1. Gas Chromatography (GC)

In GC, the components of a vaporized mixture are separated based on their differential movement rates while moving in the gaseous mobile phase through the stationary phase [10,11]. GC-MS technologies are well-suited for the analysis of small molecular metabolites (<650 Da), including acids, alcohols, hydroxyl acids, amino acids, sugars, fatty acids, sterols, catecholamines, drugs, toxins, and intermediates of the tricarboxylic acid cycle and glycolysis [12,13,14]. Chemical derivatization is often employed to make compounds volatile enough for GC [13,15]. GC-MS is optimal for determining volatile substances, such as those present in fecal samples, to study the influence of the gut microbiome on health [13]. On the other hand, the limitations of GC-MS are related to the volatility of the analytes and their stability at high temperatures [11]. Thus, non-volatile metabolites, such as peptides or many lipids, as well as thermolabile compounds, are unsuitable for GC-MS [11,13,15].

Mass spectrometry is often combined with various chromatography methods, allowing for the analysis of complex mixtures and the detection of substances present in micro- and sub-micron quantities. GC-MS combines the capabilities of gas chromatography (GC) for component separation with mass spectrometry as a detection and structural elucidation method. The most popular ionization source in GC-MS instruments is EI. Quadrupole analyzers are most commonly used in GC-MS due to their low cost and simplicity, but tandem mass spectrometers, including high-resolution instruments such as TOF or QTOF, are also employed. One advantage of the GC-MS method is its standardization with established protocols for many metabolites and the availability of large mass spectral libraries.

#### 4.1.2. Comprehensive Two-Dimensional Gas Chromatography (GC×GC)

Comprehensive two-dimensional gas chromatography (GC×GC) is an advanced modification of GC, characterized by an improved separation technique. This method implies the use of two chromatographic columns connected through a modulator, which supports sample movement from one column to another [16]. Typically, these columns have different polarities and, therefore, different stationary phases. This feature enables the separation of compounds based on several properties, most commonly molecular weight and polarity. GC×GC is becoming a notorious technique in the field of metabolic profiling due to its ability to identify far more peaks compared to one-dimensional GC analysis, resulting in an increased resolution [17]. Furthermore, this approach is suitable for the analysis of complex biological mixtures, such as human plasma. Other compounds applicable to the analysis with GC×GC are similar to those with GC. Among other notable qualities of two-dimensional GC are effective background signal removal; simplified sample preparation; the high speed of separation; and, finally, the simultaneous processing of compounds of different classes. Despite the rich abilities of the presented method, it is worth mentioning its drawbacks, which should be considered during routine analysis. One difficulty is distinguishing between compounds of similar classes on the chromatogram that all elute at the same time [18]. The other difficulty arises from the expensive running costs of GC×GC, which may not be efficient for analyzing a small number of samples.

The combination of GC×GC with MS is a new method that is growing in popularity in the metabolomics field. The most widespread combination is GC×GC with TOF MS. This method has several advantages over the common GC-MS, such as better metabolite coverage, separation peak capacity, sensitivity, and resolution [19,20]. These enhancements render GC×GC-MS a powerful tool for untargeted metabolomics, as it is capable of detecting three times the number of metabolites compared to traditional GC-MS [21]. Thanks to its separation superiority, there are many metabolites that can only be detected by GC×GC-MS and not by GC-MS, such as carbohydrates and carbohydrate-conjugate metabolites [19].

#### 4.1.3. Liquid Chromatography (LC)

The concept of LC encompasses methods for separating and analyzing substance mixtures where the mobile phase is a liquid. High-performance liquid chromatography (HPLC) is a traditional variant of LC, characterized by the utilization of small particles of sorbent (2–20 μm) and high inlet pressure (up to 500 bar) [22]. It is an automated and standardized method with good reproducibility, universality, and separation efficiency. It can rapidly and cost-effectively determine numerous compounds within a single analytical cycle [23].

A more modern variant of HPLC, involving the use of sorbents with particle diameters below 2 μm and operating at higher pressures (up to 1500 bar), is referred to as ultra-high-performance liquid chromatography (UHPLC) [22]. UHPLC offers improved separation efficiency compared to HPLC and enables high-resolution separation of complex samples. Another advantage of UHPLC is its ability to perform very fast separations with high resolution, increasing laboratory throughput (i.e., the number of analyzed samples per unit of time) while saving solvents [24,25]. The method provides high precision, reproducibility, low limits of detection, and excellent compatibility with HPLC, making it relatively easy to transfer methodologies developed for HPLC.

LC-MS combines liquid chromatography (LC) with mass spectrometry (MS). The most common strategy for studying metabolome composition involves coupling liquid chromatography with a mass spectrometer equipped with an ESI source. Coupling with MS allows for reliable compound identification, making HPLC-MS suitable for high-throughput screening of a large number of samples. HPLC-MS can handle a wide range of compounds, including both nonpolar and polar metabolites, thermally unstable and/or non-volatile compounds, as well as high-molecular-weight compounds, such as complex lipids [13,14,26]. In comparison to direct injection methods in MS, HPLC-MS provides better specificity and lower limits of detection through chromatographic separation [23]. HPLC-MS has limitations related to the solubility of the sample in the solvent or compatibility of the sample with the solvent, but overall, these limitations are less restrictive than those in GC.

#### 4.1.4. Capillary Electrophoresis (CE)

The CE separation method is based on the differential electrophoretic mobilities of charged species in capillaries. By utilizing a small capillary and subjecting charged molecules (ions) to an electric field, CE separates these molecules based on their movement towards electrodes. Capillary electrophoresis–mass spectrometry (CE-MS) combines mass spectrometry with the separation power of capillary electrophoresis [22]. CE-MS is particularly suitable for the analysis of amino acids, nucleotides, small organic acids, and sugar phosphates due to its ability to effectively profile polar and charged metabolites [27,28]. The technique is characterized by its high resolution, separation efficiency, sensitivity, analysis speed, ability to provide structural information, and low sample consumption [28]. Many studies have highlighted its capacity for analyzing volume-restricted samples, as it requires only a few nanoliters of the sample [27,28,29]. Another advantage of CE-MS, although less highlighted, is its high reproducibility, making it suitable for large-scale experiments [27,28]. In addition to these advantages, many comparable datasets have been collected using the same sample preparation, processing, and data interpretation procedures for CE-MS [28]. Despite its advantages, CE-MS has limitations, as it is not recommended for anionic metabolic profiling [27] and there is a lack of standard operating procedures and workflows for CE-MS [27].

### 4.2. Mass Spectrometry (MS)

MS is a method based on the ionization of molecules followed by the separation of the resulting positively or negatively charged ions, depending on their mass-to-charge ratio (*m*/*z*), and the detection of the separated ions [30,31]. Each formed ion is characterized by a specific *m*/*z* value, which can be accurately measured, allowing for the determination of its molecular mass [30,31,32]. Additionally, the abundances of ion signals determine the content of the components in the sample. Fragmentation of the initial molecular ions can also be initiated, resulting in a set of fragment ions. These ions represent the most stable fragments of the original molecule and provide insights into its structure. Thus, mass spectrometry is a sensitive and powerful qualitative and quantitative analytical technique for the identification and quantification of compounds in a sample.

The main components of a mass spectrometer are the ion source, which converts sample molecules into charged ions in the gas phase; the mass analyzer, which uses electric and/or magnetic fields to separate ions based on their *m*/*z* values; and the detector, which measures the signal of each ion [31].

All existing ionization techniques can be broadly divided into “soft” and “hard” methods [30,31]. This division is based on the amount of internal energy excess in the resulting molecular ion. In “soft” ionization, this excess is small, the ion fragmentation is negligible, and the ion is detected as a molecular ion. Contrarily, in “hard” ionization, a significant excess of internal energy triggers the breaking of specific chemical bonds in the original molecular ion, resulting in the appearance of various fragment ions. The main ionization methods are briefly described in the following sections.

#### 4.2.1. Electron Ionization (EI)

Electron ionization (EI), the most popular “hard” ionization method for organic molecules, was first described by Dempster in 1918 [30,31]. In electron ionization, the sample molecules in the gas phase are subjected to a stream of electrons with a specific energy (most commonly 70 eV) under low-pressure conditions. Typically, the molecule loses an electron and becomes a positively charged ion (radical cation). Molecular ions obtained under such conditions have a significant excess of excitation energy, leading to their fragmentation. EI spectra provide rich structural information and can be used as characteristic fingerprints of specific molecules [30,31]. Furthermore, since EI spectra are highly reproducible, representative libraries of such spectra have been created (at 70 eV), such as the mass spectral library of the National Institute of Standards and Technology, Wiley^®^ (John Wiley & Sons, Inc., Hoboken, NJ, USA) mass spectral databases, and others [13,15]. Researchers effectively utilize these libraries for compound identification. EI is a versatile method that allows the investigation of various classes of chemical compounds. However, this method has the disadvantage of requiring the conversion of the sample molecule into the gas phase, which may limit its applicability to polar, thermolabile, and non-volatile molecules [13,31]. Nevertheless, some of these molecules can be converted into suitable derivatives for analysis through derivatization using various reagents [10,13]. The EI source is ideally compatible with quadrupole instruments, although other types of analyzers can also be used. Mass spectrometers with EI sources are commonly coupled with gas chromatography (GC) and are typically employed in GC-MS analysis of organic molecules.

#### 4.2.2. Atmospheric Pressure Ionization (API)

Atmospheric pressure ionization (API) represents a pioneering technique that establishes a direct link between the supply of analytes in the solution phase and a mass analyzer. The prevailing method of ionization under atmospheric pressure conditions is electrospray ionization (ESI). In ESI, the sample solution is aerosolized at the tip of a slender capillary needle under high voltage (either positive or negative relative to the inlet of the mass spectrometer), inducing the formation of minute, charged droplets. As the solvent undergoes evaporation and the droplets diminish in size, ionized molecules from the sample are emitted. This method has significantly advanced the investigation of non-volatile, labile molecules, and compounds with high molecular weight. A complementary soft ionization technique to ESI is atmospheric pressure chemical ionization (APCI), often employed for analyzing polar and nonpolar compounds that exhibit poor ionization efficiency with ESI but respond favorably to chemical ionization. In the case of APCI, ions are generated in the gas phase through the application of a corona discharge, facilitating the ionization of solvent molecules and analytes within the aerosol. Subsequently, ions transitioning into the gas phase are analyzed via mass spectrometry. APCI, with its reduced susceptibility to matrix effects, including ion suppression compared to ESI, is deemed suitable for various applications, including the determination of nonpolar analytes. ESI and APCI are exceptionally mild ionization methods, characterized by minimal fragmentation of the formed ions. Although ions do not undergo fragmentation during these ionization methods, they are advantageous for collision-induced dissociation (CID), allowing the generation of MS/MS spectra on tandem instruments and aiding in the elucidation of molecular structures. The preparation of samples for ESI and APCI involves dissolving the analyte in water or a water-based mixture with polar organic solvents, typically methanol, isopropanol, or acetonitrile. Owing to their ability to directly provide ions from molecules in solution, ESI and APCI are particularly useful in liquid chromatography applications.

#### 4.2.3. Tandem Mass Spectrometry (MS/MS)

Tandem mass spectrometry resolves the issue of obtaining structural information about substances forming ions in the source, as it is a powerful analytical approach for elucidating the structure of a substance molecule and quantitatively assessing its content. The most commonly used approach for ion fragmentation in MS/MS analysis is CID [12,31,33]. In this process, selected precursor ions acquire high kinetic energy due to acceleration by electric potential. The accelerated ions then collide with neutral gas molecules in a collision cell. As a result, part of the kinetic energy is converted into internal energy, leading to the fragmentation of precursor ions and the formation of ion fragments with *m*/*z* values containing chemical information. The CID process can be performed at low and high collision energies.

Conducting tandem experiments requires the use of a combination of multiple mass analyzers or an ion trap mass spectrometer. Modern mass spectrometers employ various hybrid combinations of mass analyzers, such as triple quadrupole (QQQ), quadrupole time-of-flight (QTOF), or Orbitrap mass spectrometers. The QQQ combination allows for several modes of tandem mass spectrometry: product ion scan, precursor ion scan, neutral loss scan, selected reaction monitoring (SRM), or multiple reaction monitoring (MRM) [12,31,34]. The SRM/MRM mode is widely used for quantitative analysis as it offers high specificity and a low limit of detection. Non-target components cannot be detected using a QQQ instrument operating in SRM/MRM mode. The main drawbacks of all quadrupole instruments are their low resolution and, consequently, limited accuracy in measuring *m*/*z* for unambiguous compound identification. In contrast, QTOF and Orbitrap instruments provide accurate mass measurement, enabling the acquisition of elemental composition information for analyzed compounds [31,32]. Tandem experiments, similar to the product ion scan mode for QQQ instruments, can be conducted on QTOF or Orbitrap mass spectrometers by transitioning to MS/MS mode. The resulting high-resolution CID spectrum can then be used for compound identification. Replacing the third quadrupole in a QQQ instrument with an Orbitrap mass analyzer enables the simultaneous detection of all target product ions in a single coherent high-resolution mass analysis. This operation, termed parallel reaction monitoring (PRM), mirrors SRM scanning, with the exception that all transitions are collectively detected and separated from each other and the background in the final stage of analysis. This operational mode is referred to as parallel reaction monitoring (PRM) (source). Given that PRM tracks all transitions, there is no need to pre-determine the target transitions or pre-select them before analysis, distinguishing it from SRM/MRM.

## 5. Research Field Landscape

Metabolomics is a broad area that encompasses various approaches to metabolite profiling based on their composition. At present, six major fields of metabolomics can be defined, including lipidomics, glycomics, fluxomics, ionomics, and metallomics. Lipidomics is an area that analyzes lipids, playing one of the most significant roles in any living organism, facilitating energy metabolism and signaling. Glycomics focuses on carbohydrates and explores their role in disease progression [35]. Glycans are an integral part of the body, involved in protein clotting, participating in protein transport, and modeling the immune system, and are, therefore, of particular interest. Metabolomics also includes some other fields that are infrequently used with blood probes. Ionomics involves the analysis of the ionome, including inorganic components and trace elements that interact with living organisms [36]. Metallomics, a branch of ionomics, aims to determine the forms of metals and other metalloid elements in living systems and to establish their functions and the mechanisms of the biological processes in which they participate [37]. The most widespread analysis in these fields is microelement detection in hair samples. Fluxomics is the study of metabolic reactions or flows in biological systems [38,39]. Fluxomics evaluates the relative or absolute rate of metabolic reactions through a series of metabolic intermediates in a given metabolic pathway [40]. Fluxomics studies usually involve cellular cultures.

### 5.1. Lipidomics

Lipidomics is a field of qualitative and quantitative analysis of all known lipids, describing their properties in biological systems [41,42,43]. Lipids play one of the most significant roles in any living organism, facilitating energy metabolism and signaling processes. Lipidomics profiling is widely applied in studies of diseases based on inborn metabolic errors [44]. The methods of lipidomics enable the identification of specific lipid classes that accumulate in the body during these diseases, which could later be used as diagnostic targets. Quite recently, a new branch of lipidomics has emerged: clinical lipidomics [45,46]. This area of study considers lipid profiles in relation to the pathways and pathological processes in which they are involved. To date, many lipid classes linked to various diseases have been identified [47]. The next step in developing clinical lipidomics is to integrate it into personalized medicine. Based on current literature, the majority of lipidomics studies focus on fatty acids, oxylipins, phospholipids, glycerophospholipids, sphingolipids, and sterol lipids. All categories will be further reviewed in detail below.

#### 5.1.1. Fatty Acids

Fatty acids, fundamental constituents of lipids, are central players in human physiology, exerting a profound and expansive influence. Diverse in composition, fatty acids span various categories, including saturated, unsaturated (monounsaturated and polyunsaturated), and trans fats, each with its unique chemical structure. These molecules serve as repositories of caloric energy and as indispensable components, contributing to the structural integrity of cell membranes and serving as precursors for various bioactive compounds. Fatty acids actively participate in energy metabolism through beta-oxidation to generate adenosine triphosphate (ATP), while also modulating cell membrane fluidity and participating in essential cell signaling pathways.

However, their role extends beyond the above, since fatty acids are associated with a wide spectrum of diseases. Elevated levels of saturated and trans fats are linked to cardiovascular diseases, while polyunsaturated fatty acids, including omega-3 and omega-6 variants, assume pivotal roles in regulating inflammation and maintaining cardiovascular health. Dysregulation in fatty acid metabolism can significantly contribute to conditions such as obesity, diabetes, and metabolic syndrome [48]. Employing derivatization-enhanced separation with LC-MS, a breakthrough quantified long-chain free fatty acids (LCFFAs) in the sera of asthma patients, shedding light on non-cancer disease-associated metabolic changes [49]. The impact of ischemic stroke on plasma-free fatty acid (FFA) derivatives, including lipoxins, RevD1, and 9,13 HODE, has been identified as critical mediators with potential implications for quenching inflammation [50]. This underscores the importance of studying specific FFA derivatives in the context of ischemic stroke and highlights their potential as therapeutic targets for modulating inflammatory responses. In the realm of chronic kidney disease (CKD) progression, altered fatty acid profiles have been characterized by changes in inflammation-related pathways, decreased polyunsaturated fatty acids (PUFAs), and increased monounsaturated fatty acids (MUFAs) [51]. These findings highlight the potential role of fatty acids in the pathogenesis of CKD and may offer opportunities for novel therapeutic interventions targeting fatty acid metabolism. Furthermore, utilizing an animal model of Yersinia pestis infection, significant changes in lipid profiles and fatty acid proportions have been observed, suggesting potential implications for inflammation and oxidative stress [52].

The clinical application of fatty acids is influencing dietary guidelines by emphasizing the reduction of saturated and trans fat intake, thus directly impacting the management of cardiovascular risk [53]. Personalized nutrition plans and therapeutic interventions tailored to individuals with conditions such as diabetes and obesity significantly benefit from targeted fatty acid analysis. Omega-3 fatty acid supplementation, known for its potential cardiovascular and anti-inflammatory benefits, serves as a prime example of how fatty acid research translates into practical clinical applications, reshaping the landscape of preventive medicine and patient care.

In our array of studies reviewed, fatty acid profiling investigations are more likely to utilize MS approaches with LC [48,49,52], rather than GC [50]. We assume this is because LC, especially in its high-resolution or high-performance variations, offers much higher sensitivity and selectivity compared to GC-MS.

#### 5.1.2. Oxylipins

Oxylipins are a family of oxidized lipids derived from polyunsaturated fatty acids, such as omega-6 or omega-3. These bioactive molecules are synthesized during inflammation or infection, thus playing a role in immune defense and modulating the inflammation response. Oxylipins can be produced by almost all cell types but cannot be stored in tissues. They are formed from 20 essential amino acids, which define their pro-inflammatory or anti-inflammatory effects [54]. The concentration of oxylipins may change with an individual’s age or during certain diets. According to the study by Caligiuri et al. [55], a four-week flax seed diet had a positive impact on restoring the oxylipin profile in adults. The same study stated that the level of pro-inflammatory oxylipins increases in older individuals [55].

Since oxylipins modulate the inflammatory response, they are involved in the pathogenesis of various inflammatory diseases. In one of them, aortic dissection, serum metabolic profiling revealed 91 altered oxylipins [56]. Subsequent pathway analysis indicated that most metabolites were part of the arachidonic acid metabolism pathway. Oxylipins also modulate cardiovascular diseases by maintaining vascular homeostasis and regulating blood pressure [57]. High levels of oxylipins in serum were associated with an increased risk of acute myocardial infarction and other cardiovascular events [58,59]. Moreover, the characterization of oxylipin profiles was proposed in patients with neurophysiological diseases. In amyotrophic lateral sclerosis, major alterations were identified in linoleic acid-derived oxylipins and 5-lipoxygenase metabolites [60]. At the same time, plasma levels of 13-HODE and 9-HODE showed a positive correlation with disease continuity.

The clinical application of oxylipins mostly involves the evaluation and quantification of eicosanoids: prostaglandins, lipoxins, and other groups. Given the immune-modulating effects of these molecules, they may serve as reliable therapeutic agents. Thus, lipoxins have shown promising applications in cardiometabolic, cardiovascular, respiratory, and neurodegenerative disease management [61,62,63].

Modern equipment for oxylipin analysis includes processing with LC-MS/MS [64], GC-MS/MS [58], HPLC-MS/MS [60], and ESI/MS. Other known techniques, such as immunoassay and thin-layer chromatography, are not specific or sensitive enough for oxylipin profiling, often obstructed by the need to perform additional derivatization procedures [65].

#### 5.1.3. Phospholipids

Other notable compounds in the field of lipidomics include phospholipids. Structurally, these are polar molecules with a hydrophilic head containing a phosphate group and two hydrophobic tails, which are fatty acid derivatives. Phospholipids are major components of the plasma membrane, forming lipid bilayers in all living cells. They are involved in lipid, fatty acid, and cholesterol transport. There are two main classes of phospholipids, differing in their backbone content: glycerophospholipids and phosphosphingolipids (sphingolipids). Each class is further divided into subclasses depending on the types of lipid headgroups.

Altered concentrations of phospholipids are associated with a wide spectrum of metabolic, cardiovascular, and neurological diseases. For example, notable changes have been detected in arthritic diseases: collagen-induced arthritis [66] and rheumatoid arthritis (RA) [67]. Some of the phospholipids identified in these studies have been used as biomarkers, contributing significantly to the early diagnosis of RA [68]. Blood phospholipids have also shown utility in differential diagnostics for distinguishing between RA and Lyme arthritis [69]. Differentiation between NAFLD and NASH is also possible through analysis of phospholipid content, which was reported to be elevated in NAFLD subjects [70]. Hence, these are a few successful examples of phospholipid biomarkers for differentiating between similar disorders that can be used in diagnostics.

Due to their great biocompatibility, phospholipids have found applications in pharmaceutical developments. They are known to reduce levels of cholesterol and triglycerides and can increase levels of HDL, known as “good cholesterol”. Based on the above, medical treatment with phospholipids has shown positive effects on patients with liver diseases and metabolic disorders [71].

#### 5.1.4. Glycerophospholipids

Glycerophospholipids are molecules with a glycerol backbone. They play a significant part in forming the cell membrane and act as precursors for lipid mediators, modulating cellular responses [72]. Glycerophospholipid composition varies among cell types, enabling them to perform a wide range of functions, from signal transduction and vesicle trafficking to membrane fluidity control [72]. The most abundant glycerophospholipids in the blood are phosphatidylcholine (PC), phosphatidylethanolamine, and phosphatidylserine [73]. As will be shown, all these compounds are effective biomarkers for various diseases.

An altered metabolism of PCs has been reported to be associated with sepsis [74]. There is evidence of both increased and reduced concentrations of PC species in the blood. Upregulated levels of saturated and unsaturated PCs and LPC in plasma were identified in a comparison between healthy and septic individuals [75]. Notably, the levels of PC and lysophosphatidylcholine (LPC), a derivative of PC, displayed corresponding changes. Conversely, reduced levels of PC and LPC were found in neonates with late-onset sepsis, accompanied by necrotizing enterocolitis [76]. PCs and LPC may also be considered prognostic markers for evaluating survival risk. Low unsaturated long-chain PCs and LPC in patients with severe septic shock were associated with a three-month survival rate [77]. In addition to sepsis, diabetes mellitus is characterized by an altered glycerophospholipid profile. Major structural lipids, including LPCs and PCs, were reduced in non-obese diabetic mice studies [78]. Studies of human blood plasma yielded similar results to the animal model: PC levels are reduced in diabetic patients [79]. Glycerophospholipid levels may change in certain bacterial infections. For instance, in Lyme disease caused by *Borrelia* bacteria, LPCs are reported to increase [80]. This may be due to *Borrelia*’s ability to manipulate the host lipid metabolism, affecting structural support, immune evasion, and disease severity [81]. Another case of bacterial disease is melioidosis, which is caused by *Burkholderia pseudomallei*. High PC levels were reported in patients with this infection [82]. Further studies may draw upon the aforementioned results to properly explore host response in bacterial diseases and reveal how glycerophospholipid metabolism affects the pathogenesis of diseases. Glycerophospholipids show altered profiles in inflammation-related diseases. In some of them, glycerophospholipid levels are reduced, for instance, in blood vessel inflammation, known as Behçet’s disease [83]. This disorder is associated with a low concentration of several serum PCs. One more example is inflammation induced by drug exposure, which has an effect on LPC concentrations. In a recent study, LPCs were reported as potential biomarkers for drug-induced interstitial lung disease [84]. The selected biomarkers showed no associations with other drugs and were effectively implemented to differentiate between other lung diseases: idiopathic interstitial pneumonia and lung disease associated with connective tissue disease.

The detection and quantification of glycerophospholipids are mostly implemented by LC modifications with MS: UHPLC-MS/MS [74], UPLC-Q-TOF-MS [83], and UPLC coupled to ESI-QTOF [79]. For large clinical studies of glycerophospholipids, an accurate and fast method with application to different biological samples is needed.

#### 5.1.5. Sphingolipids

Sphingolipids are a subclass of phospholipids, containing a backbone with sphingosine bases or similar structures. The basic sphingolipids are the ceramides; all other sphingolipids are their derivatives. Sphingolipids play a role in maintaining the structure of cell membranes and act as cell signaling modulators and mediators. These molecules have emerged as key players in various diseases [85]. Numerous studies have explored alterations in sphingolipid metabolism and their associations with different pathological conditions. Recent findings have demonstrated the significance of sphingolipids as biomarkers for disease diagnosis, prognosis, and treatment response.

The diagnostic potential of sphingolipidomics has been examined in various diseases. For instance, Qu et al. [86] analyzed the plasma lipidome of patients with chronic hepatitis C virus (HCV) infection using a high-throughput lipidomic platform. The analysis revealed distinct differences in sphingolipid composition between healthy and infected individuals, and the lipid profile also correlated with the severity of intrahepatic inflammation, suggesting a potential role of sphingolipids in HCV-related digestive system disorders. Moreover, sphingolipid alterations were observed in acute respiratory distress syndrome associated with H1N1 influenza. The identified biomarkers, including lysophospholipids and sphingolipids, provided insights into the metabolic changes occurring in patients with acute respiratory distress syndrome, contributing to a better understanding of the disease’s pathogenesis [87]. Sphingolipid metabolism was also found to be affected by methamphetamine exposure, influencing gut homeostasis, serum metabolome, neurotoxicity, and behavior in mice [88]. Methamphetamine disrupted gut homeostasis, induced inflammation, and altered the serum metabolome, affecting Bacteroides-derived sphingolipids and serotonin, which correlated with the observed behavioral changes. Other studies investigated sphingolipid-related metabolic disruptions in spontaneous intracerebral hemorrhage, neuropsychiatric diseases such as schizophrenia, obesity-related diabetes risk, and experimental autoimmune encephalomyelitis, an animal model for multiple sclerosis [89]. These studies identified notable changes in sphingolipids and associated pathways, providing insights into the pathogenesis and potential diagnostic markers for these conditions. Bariatric surgeries, such as Roux-en-Y gastric bypass (RYGB) and adjustable gastric banding, were also investigated in relation to lipid metabolism [90]. Comparing the lipid profiles of obese women undergoing these surgeries, RYGB was reported to induce more alterations in lipid markers, including sphingolipids. These RYGB-specific alterations were associated with metabolic improvements, independent of weight loss, suggesting a specific effect of RYGB on sphingolipid metabolism and metabolic outcomes. Sphingolipid alterations have also been investigated in the context of sepsis treatment, the Bacille Calmette–Guérin vaccine [91], gestational diabetes mellitus [92], allergic inflammation, and coronary artery ectasia [93]. These studies have identified specific sphingolipids and other metabolites associated with disease presence, severity, treatment response, and early diagnosis.

Sphingolipid profiling involves the usage of high-throughput MS techniques: HPLC-MS [86], HPLC-MS/MS [90], UPLC-HRMS [87,93], and UPLC-MS/MS [91]. These platforms have shown good application in the detection and identification of a broad spectrum of sphingolipid classes, some of which could not be detected by the less complicated LC-MS or LC-MS/MS methods. These methods can also be modified for the simultaneous quantification of specific sphingolipid metabolites.

#### 5.1.6. Sterol Lipids

Sterol lipids unite cholesterol and cholesterol-derived metabolites. These compounds share a common feature: four linked hydrocarbon rings that could be modified by functional groups [94]. The largest number of studies in lipidomics is dedicated to the best-known animal sterol, cholesterol. Its main function lies in forming the cell membrane structure and serving as a precursor to several organic molecules, including bile acids, oxysterol, steroid hormones, and vitamin D. Cholesterol exists in two forms: low-density lipoprotein (LDL) and high-density lipoprotein (HDL). Studies of the latter make up a large area in lipidomics [95,96]. Popular science texts describe HDL as “good” cholesterol that absorbs “bad” cholesterol in the blood and carries it away. The structure of HDL includes a hydrophobic core of non-polar lipids, surrounded by a hydrophilic membrane consisting of phospholipids, free cholesterol, and apolipoproteins [97]. In human blood, HDL consists of 40% [98] proteins and 60% lipids (12% triacylglycerols, 40% cholesteryl esters, 47% phospholipids, and 1% free fatty acids) [98]. The most important processes in which HDL is involved in the human organism are lipid metabolism, lipid transport in the blood, and lipid storage in the liver.

Cholesterol in high concentrations may create fatty deposits in blood vessels, which makes it difficult for blood to go through the arteries. This complication results in the risk of cardiovascular disease development [99]. Lipidomics approaches offer a great opportunity to find new cholesterol biomarkers for diseases and evaluate their concentrations. For instance, in pre-eclampsia, a positive association with high cholesterol precursors and metabolite production was discovered [100]. Not only are high concentrations of cholesterol associated with complications; low concentrations also serve as indicators of certain developing diseases. Low levels of HDL are often associated with inflammation [101]. According to the found studies, inflammation processes affect the HDL structure, thus reducing the ability of HDL to participate in reverse cholesterol transport [102]. Nonetheless, the exact reason why inflammation decreases the level of HDL is still being explored. These findings are consistent with studies on several inflammatory diseases; for instance, in RA, cholesterol levels are decreased [67]. In the context of RA, researchers explored alterations in the lipidome and antioxidative activity of small HDL particles. Such alterations were found in all RA patients, especially in those with high levels of inflammation [103]. The functional deficiency of small, dense HDL in RA was attributed to inflammation, highlighting the impact of inflammatory processes on HDL structure and function [95]. Furthermore, during RA treatment, HDL proteome remodeling was observed, leading to changes in proteins involved in immune response, proteinase inhibition, and lipid metabolism [104]. Notably, normolipidemic RA patients with high levels of inflammation exhibited reduced antioxidative activity in small, dense HDL, suggesting functional deficiencies [95]. Other sterol lipids behave differently during inflammation. Oxysterol levels in chronic HCV infection, which is characterized by liver inflammation, were reported to be high [105]. These results suggest that elevated levels of oxysterols may be observed due to oxidative stress or inflammation in the liver and cholesterol autoxidation. Other examples of altered sterol lipid behaviors have been observed in several HIV studies. Although it is not an inflammatory disease, it causes inflammation. This is evidenced by changes in several sulfated steroids [106] and a decrease in neuroactive steroids, revealing an association with depression in HIV [107].

Modern techniques for sterol lipid profiling in blood include the use of MS: GC-MS, LC-MS, or MRM-MS [96]. Compared to previous methods for sterol identification, such as thin-layer chromatography or UV-vis spectrophotometry, the MS approach allows for differentiation between sterol lipids of similar structures [108]. For example, LC-MS/MS was utilized for the rapid quantification of hydroxysterols, primary and secondary bile acids, and their taurine and glycine conjugates [109]. Another study used the same approach for the quantification of oxysterols and suggested a method for their isolation using solid-phase extraction [110]. The method was tested on oxysterols from peripheral blood mononuclear cells in mitochondria.

### 5.2. Glycomics

Glycomics focuses on exploring carbohydrates (glycans) and their role in health and disease progression [35]. Glycans are long molecules constructed from monosaccharides linked by a chemical bond. They are involved in protein clotting, participate in lipid and protein transport, model the immune system, and thus are of interest to a broad range of researchers. Many glycomic studies are oriented toward profiling specific glycan subtypes: N-linked, O-linked, and glycans from glycolipid. Particular attention is given to the analysis of glycosylation, a post-translational modification that affects protein structure and consequently their biological activity. The glycosylation profile can be altered due to various pathophysiological processes. In this connection, new biomarkers can be detected by analyzing a “damaged” glycomic profile, taking into account how the host is responding to the disease. The analysis of serum glycans is one of the most promising areas in this regard. It is expected to help develop new methods for monitoring and diagnosing a wide range of diseases.

The largest number of studies in glycomics focus on N-glycans, or N-linked glycans. These compounds are among the abundant proteins in the blood that have undergone the process of N-glycosylation, which is the linkage of an oligosaccharide molecule to the nitrogen atom of an asparagine [111]. The most important functions of N-glycans are the control of protein folding and the control of the interaction between cells by directing migration patterns [112]. The structure of N-glycans can be altered in various pathological conditions: autoimmune, diabetic, or oncological. Moreover, mutations in the genes involved in glycosylation may result in a variety of diseases, particularly nervous system alterations [113].

N-glycans are obtained from both serum and plasma [114]. Any change in serum N-glycome may indicate alterations in two components: immune B-cells, which are immunoglobulins, or the liver. Immunoglobulin molecules have several different glycosylation sites; alterations in any of them can lead to failure in the regulation of major immune processes, for instance, glycan-protein binding [115]. IgG is a quite relevant compound for serum N-glycome profiling. There is evidence that the structure of IgG is modified by N-glycans [116,117]; therefore, IgG N-glycome profiling is a significant area of study. Another immunoglobulin, secretory IgA, also has the potential to become a frequently profiled antibody in glycomics. Some sites of this molecule are highly glycosylated, and any changes in the glycosylation process may affect the health state [118]. Consequently, new methods of IgA characterization are needed. As mentioned earlier, liver conditions may reflect changes in serum N-glycome. In the literature, IgA N-glycome signatures were investigated in patients with liver diseases related to hepatitis B virus infection [119]. It was concluded that IgA levels are relatively higher in patients with the aforementioned disease. N-glycan biomarker profiling applies not only to human blood samples. In veterinary medicine, this approach was used to search for serum biomarkers in dogs with osteoarthritis, which is the most common canine disease [120].

Biomarker discoveries have led to the creation of new analytical methods with the MS approach for N-glycan characterization [121]. MS methods are a standard technique for N-glycan analysis due to their high sensitivity, compared to other methods like exoglycosidase digestion or lectin binding. Novel MS variation techniques, such as microfluidic capillary electrophoresis, assist in profiling N-glycan isomers [122,123,124,125]. Accordingly, a novel approach that can directly measure serum N-glycans from MALDI-MS has been developed [126]. Another novel method for glycan analysis combines MS with dual isotopic labeling and fluorous solid-phase extraction [127]. These authors believe this method has great potential for biomarker discovery. N-glycans have applications in therapeutics; most of these molecules are antibodies and could be used as therapeutic proteins. In addition to that, N-glycan profiling can help determine the effect of drug or chemotherapy administration. Such investigations can evaluate the sensitivity or resistance to particular compounds and prevent drug resistance in the future.

The second large group of interest in the glycomics field is O-glycans. These compounds account for 10% of all glycans and are primarily known as mucin proteins. O-glycans undergo the O-glycosylation process, where a sugar molecule attaches to the oxygen atom of serine or threonine. The main biological functions of O-glycans include protection from proteases, influencing stable protein conformation by regulating protein folding, and immunity maintenance [128]. Studies of O-glycans involve analyzing O-glycosylation sites in various proteins. One such protein is IgA; this antibody has nine O-glycosylation sites, and alteration in any of them may trigger the progression of inflammatory and autoimmune diseases like systemic lupus erythematosus (SLE), rheumatoid arthritis (RA), Crohn’s disease, and others [129]. Different proteins with altered O-glycosylation have been identified for specific diseases. For instance, Tau and amyloid precursor proteins were modified in Alzheimer’s disease [130]. Distinct glycan isoforms of IgA were characterized in IgA nephropathy patients [131].

Proteins carrying O-glycans are often used as therapeutic proteins. For instance, it was shown that mucin O-glycans are able to inhibit the pathogenic action of *Candida albicans* in humans [132]. Another application of O-glycans is the development of therapeutic vaccines [133]. Their action is aimed at activating the immune response on truncated O-glycans. However, despite numerous attempts in this area, most such vaccines still lack prolonged T-cell immunity and are insufficient for complete tumor progression termination.

Current approaches for O-glycan profiling mostly include HPLC and capillary electrophoresis. All of them can be performed in tandem with MS methods. Yet, new techniques for more rapid and precise detection of O-glycans are being developed.

### 5.3. Amino Acids

Amino acids are organic molecules required for protein building. Their structure is made up of an amino group, an acidic carboxyl group, and a unique organic radical. Although over half a thousand amino acids are known, only 22 of them can be integrated into proteins; therefore, they are of major research interest. Amino acids are involved in numerous biological functions, including food digestion, tissue construction, enzyme production, and many others. There is a significant interest in examining individual amino acid metabolic pathways of synthesis and catabolism as indicators for successful diagnosis. The variability of amino acid metabolic pathways associated with pathological disorders can lead to the emergence of new biomarkers that could be detected and identified during chromatography and MS.

The influence of amino acids is evident in a wide range of diseases, including cardiovascular diseases (acute ischemic stroke [134], systolic heart failure [135], coronary artery ectasia [93], among several others [136]); metabolic diseases (involving supplementation of ketogenic diet [137,138] and diabetes mellitus); neurodegenerative disorders [139,140,141]; psoriasis [142]; and autoimmune diseases (SLE, RA). Here, we briefly examine all the listed categories and their involvement in amino acid metabolism. Being indicators of cellular functionality, amino acid metabolites can reflect abnormal changes in inflammatory and immune responses [143]. It is important to consider their unique correlations between disease, bacterial load, metabolomic signatures, pathways, and amino acid concentrations, as they can potentially act as biomarkers or therapeutic targets of preliminary diagnosis. Inflammation significantly changes amino acid pathways. Lustgarten et al. [138], demonstrated that inflammation can lead to the transformation of metabolic pathways through activation of the tryptophan-cleaving enzyme, indolamine 2,3-dioxygenase, resulting in fluctuations in the kynurenine-related metabolites kynurenine, kynurenic acid, and 3-hydroxykynurenine.

Amino acid variations have been noted in papers investigating other cardiovascular and heart diseases. Recent studies applied an LC-MS/MS approach to explore the associations between cardiometabolic risk factors and concentrations of plasma amino acids, methylarginines, acylcarnitines, and metabolites of tryptophan catabolism. The patient cohort included individuals with arterial hypertension, coronary heart disease, and a control group without any cardiovascular diseases. The study revealed that almost all significantly different acylcarnitines, amino acids, methylarginines, and intermediates of the kynurenine and indole tryptophan pathways were elevated.

Another investigation highlighted several elements of arginine metabolism, including arginine, agmatine, creatine, and some others, as inflammation-related biomarkers. These compounds revealed metabolic abnormalities in the systemic inflammatory processes of myocardial infarction, ulcerative colitis, and ischemic heart disease [140]. The study provided a robust and rapid approach to characterizing inflammation and immune-related amino acid metabolites, which can be prospectively used in biomarker discovery to assess inflammation and immunity states in various pathological conditions. Still, another research work provided a comparison of serum amino acid metabolites in healthy control and HIV-infected subjects [144]. These authors utilized a targeting method based on LC-MS/MS, which helped rapidly characterize immune changes and inflammation-related metabolites. Compared to traditional methods, the extraction and derivatization procedures were greatly simplified without compromising performance, while method evaluation demonstrated the efficiency and reliability of the established method.

Liao et al. [145] identified 36 amino acid biomarkers related to acute liver failure. Among the detected metabolites, 27 were shown to be significantly decreased in patients receiving Mahuang decoction therapy. Having reconstructed the pathways for these identified metabolites, the authors highlighted eight of them, which can induce the efficacy of Mahuang decoction therapy in patients with acute liver failure. These eight pathways include the TCA cycle, metabolism of retinol, tryptophan, arginine and proline, nicotinate and nicotinamide, phenylalanine, cysteine, and methionine, as well as phenylalanine, tyrosine, and tryptophan synthesis. The combined concentration parameter, calculated as [asparagic acid] + [threonine] + [tryptophan] − [histidine] − [phenylalanine], showed the strongest correlation with the number of painful joints, the number of swollen joints, and the Disease Activity Score-28 for the evaluation of the severity of RA [146]. This amino acid analysis structure and their associated metabolites offer opportunities for diagnosis, as well as for monitoring disease progression and therapy in RA.

The levels of serum aromatic amino acids (AAAs), including phenylalanine, tyrosine, and tryptophan, are reliable biomarkers that reflect the severity of pathologies of various kinds. Tryptophan is an essential amino acid required not only for protein synthesis but also for important biological functions such as stress response, sleep, mood and appetite regulation, glucose homeostasis, and immune function. Most of these functions are related to metabolic pathways involved in tryptophan catabolism. Phenylalanine is an essential aromatic amino acid that must be included in dietary proteins. When phenylalanine enters the body, it is usually converted to tyrosine, which in turn degrades to acetoacetic and fumarate [147]. An apparent increase in phenylalanine and a decrease in tyrosine levels may indicate impaired phenylalanine and tyrosine metabolism and reduced phenylalanine hydroxylation in patients with Alzheimer’s disease.

In the studies found, LC-MS/MS is a common approach to amino acid metabolite analysis. Compared to other methods, this technique shows much higher sensitivity and specificity for amino acid profiling. Recently, a group of researchers proposed an improved method for quantitative amino acid analysis with LC-MS/MS, which uses small sample volumes and has an overall low cost of sample preparation [148]. Additionally, some authors use the LC-MS/MS approach coupled with TOF MS.

## 6. Disease Study Landscape

All areas of metabolomics are successfully used in the profiling of a variety of non-cancer diseases. The integration of MS-based metabolomics has transformed our understanding of disease pathology. Through the investigation of certain blood or serum metabolites, many studies have revealed crucial insights into disease mechanisms and potential therapeutic targets. While challenges and limitations persist, ongoing advances in analytical techniques, standardization, and the exploration of emerging technologies offer exciting prospects for further advancements in the field. Against this background, our review covers some of the most socially significant non-cancer infectious and metabolic diseases: tuberculosis, sepsis, human immunodeficiency virus (HIV), diabetes, non-alcoholic fatty liver disease (NAFLD), rheumatoid arthritis (RA), systemic lupus erythematosus (SLE), and several other diseases.

### 6.1. Tuberculosis

Tuberculosis, a bacterial infection caused by *Mycobacterium tuberculosis*, primarily affects the respiratory system but can also harm vital organs, such as the kidneys, spine, or brain. Tuberculosis is spread through the air by coughing and sneezing. Without proper treatment, advanced cases of tuberculosis may lead to death, which happens in 50% of cases. Tuberculosis is present globally: a quarter of the world’s population has tuberculosis in a non-contagious form. According to the World Health Organization’s (WHO’s) annual report, tuberculosis claimed the lives of 1.6 million individuals in 2022, ranking it among the leading infectious killers worldwide [149]. The highest incidence rates are reported in Southeast Asia and African nations. Major risk factors for tuberculosis include smoking and immune deficiencies [150].

Tuberculosis manifests itself in two primary forms: latent and active. Individuals with a latent form of tuberculosis have germs in their organism, but they are not contagious to other people. This state can last for decades, often eluding detection by conventional diagnostic tests [151]. If left untreated, latent tuberculosis may progress into its active form, marked by pathogen multiplication and tissue damage, accompanied by evident clinical symptoms. The clinical presentation of tuberculosis differs, depending on the affected organ. Pulmonary tuberculosis, affecting the lungs, constitutes approximately 80% of all tuberculosis cases. In contrast, when the infection occurs outside the pulmonary parenchyma, it is termed extrapulmonary tuberculosis. With appropriate medication, patients can achieve complete recovery, termed cured tuberculosis.

Tuberculosis infection strongly calls for new biomarkers for diagnostics. The acute need for biomarkers stems from by the tremendous number of incorrectly diagnosed people. About 40% of patients with tuberculosis in middle-income countries are given a wrong diagnosis. Fortunately, the rapid development of metabolomics with MS applications makes it easier to find suitable candidates for early tuberculosis identification [152,153]. Biomarkers, detectable in both plasma and serum, are accessible in active and latent tuberculosis [154,155,156], and across extrapulmonary and pulmonary cases. Recent research highlights elevated levels of leucine and kynurenine coupled with reduced citrulline and glutamine levels in latent tuberculosis [157]. In osteoarticular tuberculosis, lipid metabolites were found to be a significant category of biomarkers [158]. Lipid metabolites, particularly glycerophospholipids, feature prominently in biomarkers associated with tuberculosis progression [79,159]. The absence of standardized laboratory criteria for cured tuberculosis often results in the early discharge of patients who still have a spreading bacterial infection. This issue underscores the need for further research into novel biomarkers. One study demonstrated that l-histidine, arachidonic acid, biliverdin, and l-cysteine-glutathione are promising markers with differential expression in cured patients [160].

Any form of tuberculosis can be treated with antibiotics. A single medication typically suffices for latent tuberculosis, while the active form requires multiple antibiotics to enhance therapeutic outcomes. Like any bacterial infection, multidrug-resistant strains of tuberculosis bacteria are emerging. Utilizing high-performance chromatography coupled with mass spectrometry has yielded encouraging results in identifying specific multidrug-resistant tuberculosis biomarkers [161]. Combining multiple metabolites in an integrated approach facilitates accurate assessment of treatment response [162], holding potential for epidemic control and therapeutic advancements.

Tuberculosis presents a serious challenge to farmers, because specific *Mycobacterium tuberculosis* strains, such as *Mycobacterium bovis*, can infect livestock, particularly cattle. Therefore, specific biomarkers are needed for the differentiation between infected and healthy animals. Successful implementation of a metabolomics approach has enabled differentiation between infected and non-infected badgers [163].

### 6.2. Sepsis

Sepsis is a serious, life-threatening condition characterized by an overwhelming bodily response to an infection of any nature, whether bacterial, viral, or fungus. The hallmark of sepsis is its spread through the blood, causing damage to organs. Sepsis could affect any organ, but more frequently it damages the lungs, kidneys, liver, cardiovascular system, and nervous system [164]. On a global scale, 50 million new cases of sepsis are reported annually in both technologically advanced and economically challenged regions [165]. This condition can develop in any individual, especially adults over 65 years old, individuals with a weak immune system, or with chronic conditions like acquired immunodeficiency syndrome (AIDS), diabetes, and cancer, as well as sepsis survivors and those recently discharged from hospitals.

One of the widely used blood biomarkers for bacterial sepsis diagnosis is procalcitonin (PCT) [166]. Its levels are low in healthy individuals, but during bacterial infection, including sepsis, the level of PCT drastically increases. The quantification of PCT in human serum is accomplished through high-resolution mass spectrometry [167]. A diagnostic approach to sepsis could implement a metabolomics approach as well. A study by Guan et al. [168] employed plasma metabolomics analysis with UPLC-Q-TOF/MS to search for potential markers of sepsis, resulting in the identification of fifty-five plasma metabolites.

Treatment of sepsis requires antibiotic administration and admission to an intensive care unit (ICU). Despite a high number of recovered cases of sepsis, about 30–40% of patients have a poor outcome. New prognostic biomarkers are needed to make correct predictions and positively change the route of medical treatment. Significantly, amino acid metabolism played a pivotal role in distinguishing outcomes over different time periods, namely 28 days, hospital stay, and 90 days [169].

Circulating histones in plasma and specific lipid classes, such as cholesterol esters and oxidized phospholipids in high-density lipoprotein, are relevant markers for predicting sepsis prognosis [170,171,172]. Among other predictive biomarkers from lipid classes are PC and LPC [74]; these lipids were selected as potential biomarkers in studies exploring sepsis progression [75,173]. Distinct metabolic alterations become evident at various stages of sepsis progression, providing crucial insights into its dynamic nature. Bacterial species are one of the most common causes of sepsis development, and they differ in healthy and septic individuals. Recent studies have emphasized the importance of exploring metabolomics profiles for predicting different sepsis types, including bacteremic sepsis in emergency room settings [174]. For instance, one of the recent studies claims that the metabolites of *Escherichia coli* in septic patients are mostly involved in amino acid metabolism, protein digestion, and absorption [175]. However, profiles of healthy and non-healthy metabolites of *Escherichia coli* share some identical pathways, for instance, amino acids, peptides, terpene glycosides, and carbohydrates. In pediatric meningococcal sepsis, blood IgG Fc glycosylation sites emerge as critical indicators correlated with disease outcomes [176].

Differentiation between survival and non-survival in septic patients is a popular field of metabolomics applications, as metabolites associated with survival in septic patients are poorly characterized. A study by Kosyakovsky et al. [177] suggests some novel metabolites associated with sepsis survival: hydroxyisobutyrate, indoleacetate, fucose, and glycolithocholate sulfate. In another study, several upregulated and downregulated metabolites in survivors and non-survivors were identified [178]. Hence, metabolites in the tricarboxylic acid (TCA) cycle, amino acids, and energy metabolism pathways were upregulated in non-survivors, while metabolites of the urea cycle and fatty acids were downregulated.

Both early-onset and late-onset sepsis play a major role in infant deaths. Explorations of plasma metabolites have shown that glutathione and tryptophan metabolic pathways are the most disrupted in infants with sepsis [179]. On the other hand, the most common risk factor for sepsis development in infants is necrotizing enterocolitis, which induces ischemic necrosis of both the large and small intestines [180]. Scientists explored the blood of preterm infants with both necrotizing enterocolitis and late-onset sepsis and found several distinguishing metabolites and proteins [181]. Another study showed that molecules identified as PCs or LPCs are both reduced in infants with late-onset sepsis, indicating their potential as biomarkers for early detection [76]. There is also a comprehensive study on changes in metabolites in neonates during a week of sepsis progression [182]. Changes were marked in terpenoid skeleton biosynthesis, pyruvate metabolism, cysteine/methionine metabolism, and ascorbic acid metabolism.

Sepsis extends its influence beyond a systemic response, impacting organs such as the heart and the brain. Metabolomic markers come to the fore in distinguishing patients with sepsis-induced cardiac dysfunction. Kynurenic acid and gluconolactone stand out as promising biomarkers for this aim [183]. Additionally, kynurenic acid and galactitol were selected as good markers to distinguish survivors and non-survivors of the above-mentioned condition. In the context of sepsis-associated encephalopathy, 4-hydroxyphenylacetic acid appears to be a promising biomarker, showcasing a correlation with the severity of consciousness disorders [184]. This biomarker holds the potential to serve as a valuable prognostic tool for patients with sepsis-associated encephalopathy. Metabolites of sepsis as a secondary condition were also explored on an example of hospital-acquired pneumonia [185], where lipids showed the biggest alterations in patients with pneumonia, which hints at their possible role in modulating the inflammatory response and detoxification.

Exploring the metabolomics biomarkers in a model of sepsis induced by *Pseudomonas aeruginosa* in burned mice provides insights into potential diagnostic markers [186]. Metabolites involved in amino acid metabolism, pyrimidine metabolism, tricarboxylic acid cycle, glutamine, glutathione metabolism, and a major component of the protein collagen show promise for early diagnosis of sepsis caused by *Pseudomonas aeruginosa*. In the expansive realm of sepsis studies, the multifaceted approaches, encompassing clinical symptoms, diagnostic biomarkers, and metabolomic signatures, collectively provide a deeper understanding of this complex condition. These findings not only contribute to refined diagnostic strategies but also pave the way for targeted therapeutic interventions, aiming to improve patient outcomes and survival rates in the face of sepsis.

### 6.3. Human Immunodeficiency Virus (HIV)

Human immunodeficiency virus (HIV), one of the most critical global health concerns, damages the immune system’s white blood cells: CD4 T-helper cells, macrophages, and dendritic cells. Damage to these cells severely weakens the immune system and makes a person more susceptible to infections, which may result in the acquisition of chronic diseases. HIV remains one of the major issues in the world. As of 2022, 39.0 million individuals are living with HIV, and 630,000 people died in that year from HIV-related diseases. Annually, approximately 1.5 million new cases of HIV emerge, underscoring the persistent urgency surrounding this viral infection. HIV exists in two primary types: HIV-1, prevalent globally, and HIV-2, predominantly confined to West Africa and its neighboring regions [187]. The intriguing contrast between these types lies in the slower progression of HIV-2, with a mortality rate approximately two-thirds lower than that of HIV-1. Both HIV types are further classified into subtypes based on their geographical and genetic features.

Any HIV infection has three stages: acute HIV infection, chronic HIV infection (or asymptomatic HIV infection), and acquired immunodeficiency syndrome (AIDS) [188]. The first stage, which develops within 2–4 weeks of infection with the virus, is marked by a high viral concentration, rapid virus replication, and the initiation of attacks on CD4 cells. This acute infection stage lasts two weeks, giving way to the chronic HIV infection stage. During this period, virus replication becomes low, eventually resulting in AIDS after a decade.

HIV infection manifests itself in altered metabolic pathways, with a notable occurrence of poor sleep in over 50% of HIV-infected patients. This phenomenon is linked to the activation of the tryptophan–kynurenine pathway, leading to the generation of toxic metabolites, subsequent apoptosis, and cognitive decline [189]. The identification of these metabolic signatures associated with HIV infection provides not only a deeper understanding of its progression but also potential avenues for targeted therapeutic interventions. Notably, about 0.5% of people with HIV infection are able to maintain a very low undetectable number of viral cells for more than a year. Usually, low rates of viral cells may be achieved after long therapy, but in these people, it happens naturally. These patients belong to the elite controller group, and the exact reasons for achieving undetectable viral loads are supposed to be either genetic changes or special adaptations of the immune system to the virus.

HIV care involves special treatment with antiretroviral therapy (ART) [190]. Its main goals are to reduce the risk of transmitting HIV to other individuals, improve the immune system, and stop virus growth. Therapy is mostly aimed at preventing AIDS [191]; however, ART introduces its own set of challenges, including impacts on metabolic profiles. The therapy induces immune cell reconstitution, leading to varying degrees of immune activation, as reflected by plasma metabolite cysteine levels [192,193]. Additionally, lipid alterations, which are a common feature in HIV patients, are further influenced by ART, especially in protease inhibitor-based therapy [106,194]. Lipids change due to the specific action of HIV: the virus promotes oxidative stress and induces inflammation, during which, blood vessels start to be affected by lipid plaques. These results indicate the need to take action to loosen up the dysregulated innate immune activation in HIV patients on ART.

ART supplementation has several risks associated with mortality or the development of serious diseases. Depression is a common condition on ART with an increased risk of mortality. It was revealed that HIV patients with depression on ART have low levels of neuroactive steroids as well as high levels of cortisol/dehydroepiandrosterone sulfate ratios [107]. Thereby, the neuroactive steroid pathway is a promising target for the treatment of individuals with depression induced by ART. Another risk for patients on ART is the development of cardiovascular diseases. One marker of their progression is carotid intima-media thickness, the enlargement of the middle layer of the arteries. Serum metabolites from tryptophan catabolism showed a positive association with this feature [195]. A quite common cardiovascular disease in HIV-infected patients is coronary artery disease. Plasma biomarkers that are reported to be increased in coronary artery disease are short-chain dicarboxylacylcarnitines and glutamine/valine [196]. Remarkably, the study did not involve people on ART. Another cardiovascular disease that is a risk factor for HIV patients is left ventricular diastolic dysfunction, which is characterized by the stiffening of the heart. Metabolomic profiling of plasma samples of HIV and healthy women with and without left ventricular diastolic dysfunction revealed numerous significant biomarkers [197]. Metabolites of glycerophospholipid metabolism and fatty acid oxidation pathways in HIV patients have shown a good correlation with left ventricular diastolic dysfunction. The revealed altered metabolites in all observed cardiac dysfunctions offer new targets to prevent various cardiovascular diseases in patients with HIV.

HIV-infected patients may acquire comorbidities, some of which are of cardiometabolic nature. Metabolic syndrome often indicates the development of such comorbidities. Long-term antiretroviral therapy on its part has a significant impact on metabolomic alterations. The plasma metabolome was explored to analyze the effects of long-term ART in patients with HIV and metabolic syndrome. The analysis showed that compounds of amino acid metabolism and glutamate pathways are the main recovering pathways in such patients [198]. Apart from comorbidities, patients with HIV may acquire opportunistic infections, for example, hepatitis C. The plasma metabolome of individuals co-infected with both HIV and hepatitis C and with HCV-infected alone were compared [199]. Co-infected individuals were reported to have a decrease in glutamine and an increase in glutamic acid, arachidonic acid, and its derivatives. In addition, co-infected patients with and without liver disease progression were compared. Several metabolic pathways, including phenylalanine, tyrosine, and tryptophan biosynthesis pathways marked by an increased level of tyrosine, were revealed to be altered the most in patients with liver disease progression. Another example of an opportunistic infection is chronic obstructive pulmonary disease. Individuals with this disease and HIV were revealed to have significantly different sphingolipid profiles [200]. Particular attention in studies is given to the rarest group of individuals, which have a low undetectable number of viral cells, the elite controllers. The metabolic and lipid plasma features of these patients showed that the main metabolites associated with such strong maintenance of low virus load belong to the tricarboxylic acid cycle [201]. An increased level of α-ketoglutaric acid, which activates the mammalian target of the rapamycin pathway, is a main marker of long-term HIV control.

The multifaceted nature of HIV, spanning virological, immunological, and metabolic aspects, mandates a comprehensive approach for effective management and treatment. Ongoing advancement in diagnostics, therapeutic strategies, and metabolic profiling continues to shape our understanding of HIV, refining patient care toward a future where this global health challenge is met with increasingly effective interventions.

### 6.4. Diabetes

Diabetes mellitus is a group of chronic, metabolic diseases affecting over 537 million people worldwide [202]. Characterized by elevated blood glucose levels, this condition leads to severe damage to vital organs, including blood vessels, eyes, kidneys, nerves, and the heart [203,204]. In 2019, the WHO experts proposed a new classification of diabetes mellitus, considering both etiological factors and determining appropriate treatment [204]. At present, diabetes is categorized into six major types: type 1 diabetes (T1D), type 2 diabetes (T2D), hybrid forms of diabetes, other specific types, unclassified diabetes, and hyperglycemia first detected during pregnancy (gestational diabetes mellitus (GDM) and T1D and T2D, first diagnosed during pregnancy) [204,205]. The overwhelming majority of cases of diabetes can be attributed to one of two broad categories: T1D mellitus, which is caused by β-cell destruction and absolute insulin deficiency, or T2D mellitus, which is characterized by the presence of peripheral insulin resistance and relative insulin deficiency [206,207,208], although some cases are difficult to classify.

According to the American Diabetes Association and the WHO, diabetes may currently be diagnosed based on plasma glucose or glycated hemoglobin levels [204,209]. Advances in metabolomics offer a promising avenue to enhance predictive potential and diagnostic efficiency. Van et al. revealed a positive correlation between plasma advanced glycation end products, pentosidine, and Nϵ-(carboxymethyl)lysine, and T1D [210]. A powerful T1D-prediction model based on a set of cord serum concentrations of seven lipid metabolites was proposed by Orevsivc et al. [211]. Decreased levels of cord-blood PCs and phosphatidylethanolamines in children who were diagnosed with T1D before the age of four were also observed by La Torre et al. [212].

Concerning T2D, a higher serum indolepropionic acid level, a gut microbiota metabolite, displayed a negative association with the incidence of T2D [213]. Glycated lysine-141 of haptoglobin was proposed by Spiller et al. for enhancing diabetes diagnostic power [214]. The set of glycated lysine-141 of haptoglobin and HbA1C provided a sensitivity of 94%, a specificity of 98%, and an accuracy of 96% for T2D identification. The combination of glycated lysine-141 of haptoglobin and plasma-free fatty acids also allowed T2D diagnosis to be ameliorated [215]. The estimation of multiple glycation sites on plasma proteins was proposed as a tool for early diagnosis of T2D and adequate glycemic control [216].

As a final note on the issue of GDM diagnosis, researchers have observed disturbances in purine degradation, in metabolites involved in insulin resistance, and fatty acid oxidation have been noted. Hu et al. proposed an optimal GDM prediction model composed of glucose, uric acid, DL 11:0-iso2, L-phenylalanine, and direct bilirubin levels measured by UHPLC-MS/MS in the first trimester of pregnancy [217]. Similar alterations in metabolic pathways, including pyrimidine/purine derivatives involved in uric acid metabolism, carboxylic acids, fatty acylcarnitines, and SM in women at risk of GDM in the first trimester were noted by McMichael et al. [218]. However, their proposed panel with enhanced diagnostic accuracy included SM 14:0, hypoxanthine, alpha-hydroxybutyrate, and xanthine. Isovalerylcarnitine (C5) and tiglylcarnitine (C5:1) were also suggested by Razo-Azamar et al. [219] to be screened for early identification of pregnant women who will later develop GDM. Alternatively, serum amino acids (serine, proline, leucine/isoleucine, glutamic acid, tyrosine, and ornithine), a lysophosphatidylcholine (LysoPC(20:4)), and uric acid were observed to alter prior to GDM onset and during the period from the first to the second trimester of pregnancy [220]. Glycerophospholipid metabolism, linoleic acid metabolism, and D-arginine and D-ornithine metabolism were also proposed to be the main metabolic pathways disturbed by GDM in the second and third gestational trimesters [221]. These facts may also have clinical importance and facilitate early prediction of GDM. Diboun et al. suggested plasma glutamate measured in the second trimester as the best predictor of GDM, and a set of phosphatidylcholine diacyl C40:2, arachidonic acid, glycochenodeoxycholic acid, and phosphatidylcholine acyl-alkyl C34:3 as the best marker of GDM + T2D in pregnant women in the second trimester [222]. The diagnosis of GDM in both South Asian and white European women can be determined through an assessment of several fatty acids, together with some clinical and individual figures [223]. These fatty acids include total fatty acids, 18:2 linoleic acid, total monounsaturated fatty acids, total saturated fatty acids, lactate, and total esterified cholesterol. Among clinical features, age and body mass index were considered. Another study revealed that pregnant women with pre-existing diabetes are characterized by enhanced plasma levels of specific F2-isoprostanes in the first trimester [224]. This finding could be helpful for the detection of undiagnosed women at risk.

Metabolomics plays a pivotal role in assessing the efficacy of therapeutic approaches for managing diabetes. Utilizing UPLC-MS, baseline levels of tryptophan, bilirubin, and indoxyl sulfate, along with six-month post-surgery levels of free fatty acids including FFA 16:0, FFA 18:3, FFA 17:2, and hippuric acid, have been identified as robust predictors for the suitability and effectiveness of RYGB in T2D patients [225]. Lipidomics and metabolomics have revealed some benefits of activity in comparison with uninterrupted sitting for adults with T2D [226]. Grace et al. concluded that there are postprandial reductions in lipids associated with inflammation and increased concentrations of lipids associated with antioxidant capacity in individuals who exercise. A strong and independent positive correlation was found between the plasma lysophosphatidylinositol lipids and insulin secretion. On the contrary, dihydroceramide was negatively correlated with insulin sensitivity, while phosphatidylethanolamine and its vinyl ether-linked (plasmalogen) derivatives correlated negatively with % body fat in obese people [227]. These findings provide a new approach to both T2D early diagnosis and the estimation of the efficacy of diabetes management.

Metabolomics may be useful for screening the signs of diabetes complications, such as cardiovascular disease, kidney failure, and retinopathy. Thus, it was shown that plasma-free fatty acids with different carbon chain lengths and unsaturation were significantly upregulated in T2D patients complicated by coronary heart disease [228]. Seven lipid species, consisting of alkylphosphatidylcholine [PC(O-36:1)], cholesteryl ester, alkylphosphatidylethanolamine [PE(O-36:4)], phosphatidylcholine [PC(28:0) and PC(35:4)], and lysophosphatidylcholine [LPC(20:0) and LPC(18:2)] were identified as predictors for future cardiovascular events, and only four (alkylphosphatidylcholines PC(O-36:1) and PC(O-36:5); a diacylglycerol, DG(16:0_22:5); and a sphingomyelin, SM(34:1)) for cardiovascular death [229]. Pentosidine, a plasma marker, was correlated with coronary artery calcification score in T1D patients, serving as an early biomarker for cardiovascular disease [210]. Low circulating plasma levels of tyrosine and alanine were identified as markers for microvascular risk in T2D individuals, while decreased leucine, histidine, and valine correlated with higher mortality risk [230]. Fatty acid biosynthesis was associated with T2D complications, and specific metabolites were found to be elevated in patients with both retinal and renal complications of T2D [231].

In T2D-induced diabetic retinopathy (DR), 10 metabolites were identified as discriminators between pre-clinical, non-proliferative diabetic retinopathy, and proliferative diabetic retinopathy (PDR) [232]. Glutamic acid and glutamine emerged as reliable biomarkers for DR progression, with the glutamine/glutamic acid ratio offering improved recognition. Additionally, various metabolites, including arginine, citrulline, pseudouridine, and several others, were associated with DR in T2D patients [233,234,235,236,237,238]. Zhu et al. identified fumaric acid, uridine, acetic acid, and cytidine as potential biomarkers for PDR in comparison to long-lasting T2D without DR [239]. A cross-sectional study performed by Curovic et al. demonstrated that 2,4-dihydroxybutyric acid, 3,4-dihydroxybutyric acid, ribonic acid, ribitol, and the triglycerides 50:1 and 50:2 significantly correlated to the DR stage in patients with T1D. An increasing level of 3,4-dihydroxybutyric acid was identified as a risk marker for the progression of DR in longitudinal research [240]. The combination of amino acids and derivatives, monosaccharides, organic acids, and uremic toxins showed high performance for early diagnosis of diabetic nephropathy in T1D patients [241]. Haukka et al. revealed plasma erythritol, 3-phenylpropionate, and N-trimethyl-5-aminovalerate as the best set for the prediction of microalbuminuria [242], which is considered the first sign of kidney failure.

In T2D patients with diabetic kidney disease (DKD), disturbed pathways include cysteine and methionine metabolism [243], galactose and glycerolipid metabolic pathways [244], and taurine and hypotaurine metabolism [243]. Specific metabolites, such as glycerol-3-galactoside, were identified as predictors for DKD [244], as well as c-glycosyl tryptophan, pseudouridine, N-acetylthreonine [245], plasma histidine, and valine [246]. DKD progression may be estimated with the combination of α2-macroglobulin, cathepsin D, and CD324 [244]. Meanwhile, sphingomyelin (d18:1/24:0) was associated with a lower risk of albuminuria progression [247]. Longitudinally, the PC and several sphingomyelin species were revealed to be associated with a lower risk of the combined renal endpoint [247]. Utilizing the GC×GC-TOFMS approach, significant metabolites of sugars, sugar alcohols, amino acids, and free fatty acids were characterized in diabetes mellitus patients with kidney failure [19]. Some of these metabolites were associated with alterations in galactose metabolism and the polyol pathway.

A signature panel composed of 20 metabolites was developed by Lai and colleagues for the prediction of future T2D as a complication of GDM [248]. Moreover, elevated levels of hexoses and decreased free carnitine, acylcarnitines, long-chain non-esterified fatty acids, and phospholipids were also revealed in the cord blood of offspring born to GDM mothers [249], which confirms the transgenerational cycle of diabetes. Metabolomics also identified biomarkers for pre-term delivery and provided insights into the transgenerational cycle of diabetes [222].

Thus, metabolomics can provide a useful and promising tool not just for revealing the diabetes pathophysiological pathways, but also for prediction and early diagnosis, increasing the diagnostic power, assessing the efficiency of diabetes management, and identifying biomarkers of severe complications.

### 6.5. Non-Alcoholic Fatty Liver Disease (NAFLD)

Non-alcoholic fatty liver disease (NAFLD) is characterized by an abnormal buildup of fat in the liver, known as hepatic steatosis. It manifests itself in the absence of secondary causes such as excessive alcohol consumption, viral hepatitis, or drug-induced liver fat accumulation [250]. According to recent studies, NAFLD is a leading cause of chronic liver disease globally [251]. It is an increasingly common disease: studies with a total sample size of 8,515,431 patients from 22 countries have shown that the global prevalence of NAFLD is 25.24% with the highest prevalence in the Middle East and South America and the lowest in Africa [251]. This can be explained by its risk factors, as NAFLD is often associated with an unhealthy lifestyle, changes which are proven to lower transaminase levels and ameliorate NAFLD. Other conditions that are closely related to NAFLD include metabolic syndrome, obesity, type 2 diabetes mellitus, and dyslipidemia [250].

There are two types of NAFLD; they include non-malignant disease of nonalcoholic fatty liver (NAFL) and a more severe condition of nonalcoholic steatohepatitis (NASH). NAFL is a form of NAFLD characterized by hepatic fat accumulation with minimal or zero concurrent inflammation or liver damage. In NASH, in addition to fatty liver, there are also signs of inflammation of the liver and apoptosis. The inflammation and liver damage in NASH can cause fibrosis and may lead to cirrhosis, marked by permanent liver scarring [250].

At present, the diagnosis of NAFL and NASH is usually confirmed by liver biopsy [252], which has motivated researchers to look for other biomarkers and noninvasive detection methods. One such approach is OxNASH, a scoring method based on mass spectrometric profiling of oxidized lipid products, which correlates with histologic features of NASH [106,253]. Another approach to distinguish NAFL and NASH is by accessing hepatocellular ballooning, which is hepatocyte enlargement and the appearance of pale staining cytoplasm in the cells. Hepatocellular ballooning was proven to be a feature of NASH [254], and it positively correlates with PC and negatively correlates with LPC or lysophosphatidylethanolamine [255], offering a non-invasive diagnostic option. Researchers have explored various potential NAFLD biomarkers such as perturbations in one-carbon metabolism, mitochondrial dysfunction, and increased oxidative stress [256]. Studies have reported a significant decrease in the ratio of stearic acid to oleic acid, high levels of elongase-5 enzymatic activity [257], and high levels of homocysteine and cysteine and defective hepatic sulfur metabolism [258].

There are studies that focused on NASH specifically, showing that patients with NASH show some elevated lipids such as PC, sphingomyelin, and phosphatidylethanolamine [70]. Biomarkers of NASH also include upregulated hepatic AMPK protein and co-activated pathways of lipid synthesis and degradation [259]. Other useful markers are sphingolipids and changes in the concentration and quality of oxysterols, which can be used as non-invasive biomarkers in the categorization of NAFLD and as markers of transition from NAFL to NASH [70,260,261].

Besides the well-known risk factors such as obesity and dyslipidemia, recent studies have proven that chronic psychological distress [262], high perfluoroalkyl substances [263], and benzo[a]pyrene exposure [264] can also contribute to NAFLD initiation and progression. Currently, there is no medicine that can treat NAFLD directly; however, there are other methods for its management and resolution [265]. Predominantly, weight loss strategies [266], including bariatric or metabolic surgery in severe cases, are the most frequently employed interventions [265]. Recent studies have also shown the potential of PC for ameliorating lipid metabolism and inflammation in NAFLD subjects since it is highly correlated with ALT, TAG, HDL-C, IL-1β, and TNF-α concentrations [267]. Other solutions include treatment with vitamin E [250], increase of caffeine [268] and polyphenols [269] consumption, and the use of pioglitazone [250] or glucagon-like peptide-1 [270].

### 6.6. Rheumatoid Arthritis (RA)

Rheumatoid arthritis (RA) is a chronic inflammatory disorder that brings pain and stiffness to the joints [271]. Without timely treatment, RA may spread to other organs as well and cause inflammation. Most frequently, it affects the skin, heart, blood, eyes, or lungs. This disease is characterized as long-term; it has periods of severe and tolerable pain and currently lacks a definitive cure. Judging by the 2019 WHO report, 18 million people worldwide were living with established RA. The disease mostly affects women [272]: 70% of RA patients are female, and of these, 55% are older than 55 years.

The precise cause of RA remains uncertain, but scientists believe that genetic and environmental factors are the key causes. The molecular mechanism of RA development involves immune cells’ action against healthy joints, which results in inflammation. A primary diagnosis of RA is established based on symptoms, but an accurate diagnosis may also be achieved through specific blood biomarkers. Among the various screening biomarkers of RA, researchers identified amino acids, which proved to be associated with inflammation and steroid hormone biosynthesis [120]. The most specific serum biomarkers for RA are involved in inflammation injury, amino acid metabolism, oxidative stress, and phospholipid metabolism [67]. Reduced pathways of amino acids and glucose, along with increased fatty acids and cholesterol biosynthesis, were revealed by GC-MS of serum [273]. Non-esterified fatty acids showed a non-typical profile in RA as well. A study by Rodríguez-Carrio et al. [274] suggests that these compounds may affect RA pathogenesis due to their ability to modulate CD4+ T-cells function. According to the results, the serum non-esterified fatty acids’ profile in RA had an association with an enhanced T helper 1 cell response and an aggressive form of disease. Another set of visible changes was detected in the arachidonic acid, cyclooxygenase, and lipoxygenase pathways in rat plasma with UPLC-MS/MS [275]. The results showed increased metabolites of the cyclooxygenase and lipoxygenase pathways and arachidonic acid. As was demonstrated by Surowiec et al. [68], it is possible to find distinct metabolites of RA development years before its onset by means of LC-MS. Plasma metabolite profiling of individuals who later developed RA showed upregulation in the following lipids and small molecules: lysophospatidylcholines and tryptophan metabolism, perturbation of fatty acid β-oxidation, and an increase in oxidative stress.

RA is just one member of a group of more than 100 arthritic diseases. Distinguishing between all of them may be a hard task due to the similarity of their symptoms. Metabolomics profiling is one possible way to facilitate the differentiation of RA from other arthritis types. For instance, analyzing the phospholipid profile differentiates RA from Lyme arthritis [69]. Another comprehensive study reported unique metabolic signatures of four arthritic types, including RA [276]. These results prove that metabolomic phenotyping may be used as an effective diagnostic tool in RA and other arthritic diseases, including osteoarthritis, ankylosing spondylitis, and gout.

### 6.7. Systemic Lupus Erythematosus (SLE)

Systemic lupus erythematosus (SLE), commonly referred to as lupus, is a chronic autoimmune disease characterized by the corruption of immune cells, leading them to attack healthy tissues. This aberrant behavior triggers inflammation, which spreads to the skin, kidneys, joints, blood, lungs, and brain. In some cases, inflammation acquires a permanent state. While SLE can occur in individuals aged 15 to 45, statistics indicate that approximately 90% of SLE patients are young women [277]. Geographically, SLE displays a higher prevalence in the USA and rural regions [278]; ethnically, non-white individuals have a higher risk of SLE development. Due to a wide range of clinical manifestations and simultaneous impact on multiple organs, SLE is often mistaken for other diseases. Among the common clinical signs of SLE, skin redness is the most prominent, often presented as a “butterfly rash”, which is a persistent redness on the nose and cheeks. Additionally, SLE is associated with kidney and liver inflammation, chest pain, fever, fatigue, painful joints, and hair loss. The cause of SLE is still not established. Researchers consider it to be the result of a complex interplay between genetic and environmental factors. The triggers for SLE activation commonly include exposure to sunlight, infections, or certain medications.

SLE belongs to the broader category of lupus erythematosus diseases. It represents two major groups: manifested spontaneously, and those induced by medication. “Spontaneous” lupus types include SLE and chronic cutaneous lupus erythematosus. The latter primarily affects the skin without involving other organs, and is observed in approximately 10% of lupus cases. In contrast, drug-induced lupus erythematosus, occurring in 6% to 12% of individuals during drug exposure, can appear with symptoms similar to those of SLE, including joint pain, fatigue, fever, and inflammation. It may develop within the first weeks or months of regular medicine supplementation.

The diagnosis of SLE is challenging due to its multi-organ involvement. Primary diagnostics include visual observation, specifically searching for skin rashes and laboratory blood tests. A reduced red blood cell count is a notable marker of SLE progression. In cases where SLE affects the kidneys and liver, a biopsy of these organs may be necessary for diagnosis confirmation. At the moment, metabolomics biomarker profiling presents a promising strategy for detecting lupus disease. Key SLE plasma biomarkers have been associated with dysregulation of fatty acid [279,280], amino acids [281], and phospholipids metabolism [279], as well as disruption of tricarboxylic acid cycle [282].

Biomarkers specific to clinical manifestations may not always indicate the particular subtype of lupus, as they may present with similar clinical manifestations, necessitating reliable biomarkers to distinguish between them. Notably, lupus nephritis (LN), a common and severe complication of SLE occurring in approximately 40% of SLE patients [283], can be differentiated from SLE based on distinct serum oxylipin profiles. These profiles include several polyunsaturated fatty acids, such as arachidonic acid, linoleic acid, docosahexaenoic acid, eicosapentaenoic acid, and dihomo-γ-linolenic acid [284]. Furthermore, glycerophospholipid metabolism changes have been associated with the progression of SLE to LN. LN patients exhibit increased levels of glycocholic acid metabolites [285]. Collectively, these LN metabolites suggest involvement in inflammation, oxidative stress, and phospholipid metabolism [286]. Also, novel serum biomarkers for LN were discovered by taking into account the role of autoimmune-mediated inflammation [287].

Metabolomics profiling of plasma and serum biomarkers holds promise for monitoring lupus progression, assessing disease activity, and predicting risk states. Plasma biomarkers have also been studied during lupus progression, with research indicating that long-chain fatty acids decrease while medium-chain and free fatty acids increase as SLE advances [288]. In terms of predicting risk states, individuals with SLE face a higher risk of adverse pregnancy outcomes, a phenomenon explored through plasma metabolomics, which has led to the identification of novel biomarkers for this condition [289,290]. Such investigations may have good application in managing high-risk patients in the future.

### 6.8. Other Diseases

Polycystic ovary syndrome (PCOS) is a hormonal condition that affects 10% of women of reproductive age. This condition is caused by an imbalance of reproductive hormones, which affects ovarian function and leads to missed or irregular menstrual periods. In the studies of PCOS, which utilize metabolomics approaches, there is a noticeable trend toward the hormone profile characterization, and a search for novel metabolic biomarkers. Steroid profile characterization in women with PCOS with an MS approach was performed by Pasquali et al. [291]. The study revealed that PCOS status could be defined by incorporating several steroid concentrations, which can also accurately detect hyperandrogenemia, an increased production of androgen hormones. The result suggests that hyperandrogenism cannot be defined by interchanging hirsutism and high androgen levels. Several other hormones were revealed to be different in hyperandrogenism and non-hyperandrogenism women with PCOS, offering new insights into PCOS pathology [292]. Serum metabolic characteristics of PCOS performed by untargeted metabolite profiling were identified in several studies of adolescent and adult women [293,294,295,296,297]. In some studies, attention was paid to a certain class of metabolites rather than all categories at once. Hence, the sphingolipid profile was characterized in different types of PCOS (with insulin resistance or/and obesity) [298]. Sphingomyelin species with long saturated acyl chains were selected as potential biomarkers of PCOS. These works utilized different MS approaches: UPLC-QTOF-MS [294,295,297], Q-TOF LC/MS [293], and LC-ESI-TOF/MS [296].

Metabolomics profiling methods are widely applied to cardiovascular diseases. In the literature reviewed, we found studies on acute aortic dissection, myocardial infarction, coronary artery disease, blockade of heart vessel syndrome, resistant hypertension, systolic heart failure, and remote ischemic preconditioning. Here, we briefly discuss the application of metabolomics methods to the profiling of some of these diseases. Take acute aortic dissection, for example, one of the most common disorders affecting the aorta. It happens when there is a tear in the inner layer of the aorta, which causes the dissection of its inner and middle layers. Being an inflammation disease, acute aortic dissection is influenced by oxylipins, which modulate inflammation responses, as was noted earlier. According to recent studies, the levels of serum oxylipins in patients with acute aortic dissection were significantly altered [56].

The second broadly examined cardiovascular disease is myocardial infarction, which occurs when a part of the heart muscle dies due to insufficient blood flow. The most important question raised in myocardial infarction profiling through the means of metabolomics with MS is the identification of novel prognostic and risk-evaluation indicators. Kynurenine, a tryptophan metabolite, was significantly changed in patients with ST-acute myocardial infarction and, therefore, was suggested as a reliable marker [299]. Another predictive biomarker of risk evaluation for ST-elevation myocardial infarction and non-ST-elevation myocardial infarction is lysophospholipid [300]. Another study took plasma fatty acids and oxylipins into consideration in the task of myocardial infection risk evaluation [58]. Associations were found between several metabolites of these classes, but more comprehensive research is needed.

Heart failure (HF), a frequent complication of myocardial infarction, is also in the spotlight of metabolomics profiling. HF occurs when a heart does not pump blood properly, for the reason that the heart becomes too stiff or weak. Using a metabolomics approach with MS, scientists took a look at some metabolites that change during HF and evaluated them in individuals with HF risk for prognostic value. Cheng et al. [301] conducted the aforementioned procedure with phenylalanine, which is elevated in HF patients. For target human plasma profiling in HF samples with LC-MS/MS, an advanced protocol was created by Chan et al. [302]. The proposed method includes the profiling of 19 main metabolites, which were chosen in accordance with their biological relevance to HF. Modern studies with metabolic profiling of cardiovascular diseases utilize the following MS methods: LC-MS/MS [56], UPLC/Q-TOF [299], and GC-MS/MS [58].

Another application of metabolomics profiling was found with reference to a variety of neurodegenerative diseases and disorders, among which are Alzheimer’s disease (AD), dementia, stress disorders, schizophrenia, and sclerosis. The largest number of these studies are devoted to AD. This disease affects the brain, which results in issues with memory, thinking, and behavior. Over time, the brains of patients with AD shrink, and body functions fade, leading to death. Although the main cause of AD development has not yet been established, it is believed to be associated with amyloid plaques and the loss of brain neuronal connections. There are several risks that may provoke AD progression: head injury, high blood pressure, and clinical depression. Metabolic analysis is intended to find differently changed metabolites, which may reveal biochemical changes, and find new targets for the therapy of neurodegenerative disorders. The results of serum profiling have shown that tryptophan pathway metabolites are reduced in AD patients [141], and the glutamic acid is elevated in individuals with AD, as was revealed by the GC×GC–TOFMS approach [303]. One of the markers of AD and other diseases of this type is the plasma neurofilament light chain. The associations between this compound and some differential metabolites in AD were explored in a work by Chatterjee et al. [147]. These associations could eventually be used as prognostic markers for several other neurodegenerative diseases. Other reliable biomarkers have shown involvement in AD pathology, such as intracellular neurofibrillary tangles, extraneuronal senile plaques, neuronal and axonal degeneration, inflammation, and oxidative stress [304]. These biomarkers are a good target for the identification of disease stages as well as for predicting prognosis. With time, AD may develop into dementia; the search for potential markers for indicating its progression is a highly relevant area of study. Here, we mention one study, in which the exploration of plasma samples from AD patients provided several dementia biomarkers [305]. The changes found in the study are related to the lipid profile and feature a high accumulation of hexacosanoic acid.

One more area where the metabolomics approach has been found to be used is schizophrenia, a mental disorder that causes issues with the psyche, accompanied by episodes of psychosis. The lipidomic profile appears to be one of the main interests in profiling the blood of patients with schizophrenia [306]. The alterations resulted in myelination abnormalities, cognitive deficits, oxidative stress abnormalities, and inflammation. According to scientists, omega-3 could be used as a supplement in schizophrenia cases, as it has good tolerability and acceptability. Serum amino acid and acylcarnitine levels were also altered in schizophrenia patients, as reported by Mednova et al. [139]. The studies in this area used GC×GC–TOFMS [303], UHPLC-QTOF-MS, and LC-ESI-MS/MS approaches.

## 7. Conclusions and Future Perspectives

A comprehensive MS analysis, supplemented with a metabolomics approach, is a rapidly evolving and crucial methodology in blood profiling for non-cancer diseases. Looking back at the presented studies, we can conclude that MS indeed serves as a powerful analytical technique for identifying and quantifying metabolites of various natures. The predominant number of studies is focused on the search for novel biomarkers, which are viewed from different perspectives depending on the disease under consideration. As demonstrated in numerous studies, biomarkers are crucial for early detection, diagnosis, and monitoring the progression of the disease.

Biomarkers play a critical role in broad-spectrum diseases like SLE and RA, as they prevent misdiagnosis of similar symptomatic subtypes. Thus, different lipid classes have demonstrated their application in differentiating between various types of these diseases: phospholipids in RA and oxylipins in SLE. One of the advantages of using metabolomic biomarkers in clinical practice is that metabolic biomarkers offer a noninvasive method for confirming diagnoses, minimizing the risk of organ damage, and providing a less stressful experience for patients. This idea is especially relevant for diseases where only a biopsy is used for diagnosis confirmation, as reviewed in NAFLD.

A promising perspective involves identifying early metabolic biomarkers through a comprehensive analysis of associated metabolic pathways. As exposed in the article, potential biomarkers have already been discovered for the early detection of diseases such as tuberculosis, sepsis, RA, and diabetes, suggesting avenues for preventing severe disease progression. Another perspective is using novel biomarkers to help evaluate the disease state and optimize treatment procedures. That said, it is important to remember that therapeutic drugs also modulate the metabolome and could carry side effects that may alter vital metabolic pathways. On this matter, studies devoted to analyzing the effects of medical drugs on the metabolic profile occupy a special place in this field. This knowledge can aid in the optimization of existing therapies across various spectrums of non-cancer diseases. Overall, the analysis of the metabolomics profile has the potential to facilitate personalized medicine approaches. By looking at an individual’s metabolomic profile and taking into account their unique characteristics, clinicians can develop a specific treatment with a more effective outcome. As technological advancements continue, metabolomic profiling increases its role in clinical practice and in developing novel therapeutic approaches.

While these advancements are commendable, translating metabolomics findings into clinical practice remains a notable challenge. Bridging this gap requires integrating metabolomics-based biomarkers into clinical settings, developing standardized protocols, and considering regulatory frameworks. This transition is crucial for enabling more accurate and timely diagnoses and personalized therapeutic interventions. Recognizing the importance of closing this gap, further research and collaboration are imperative for the future of metabolomics in healthcare. Ongoing innovation and advancement underscore the need for investigations addressing these gaps, ultimately leading to more precise diagnostics and tailored therapeutic interventions. This systematic review reinforces the persisting requirement for research and collaboration to unlock the full potential of MS-based blood metabolomics in the context of non-cancer diseases and to address the critical challenge of translating these findings into practical clinical solutions.

## Figures and Tables

**Figure 1 metabolites-14-00054-f001:**
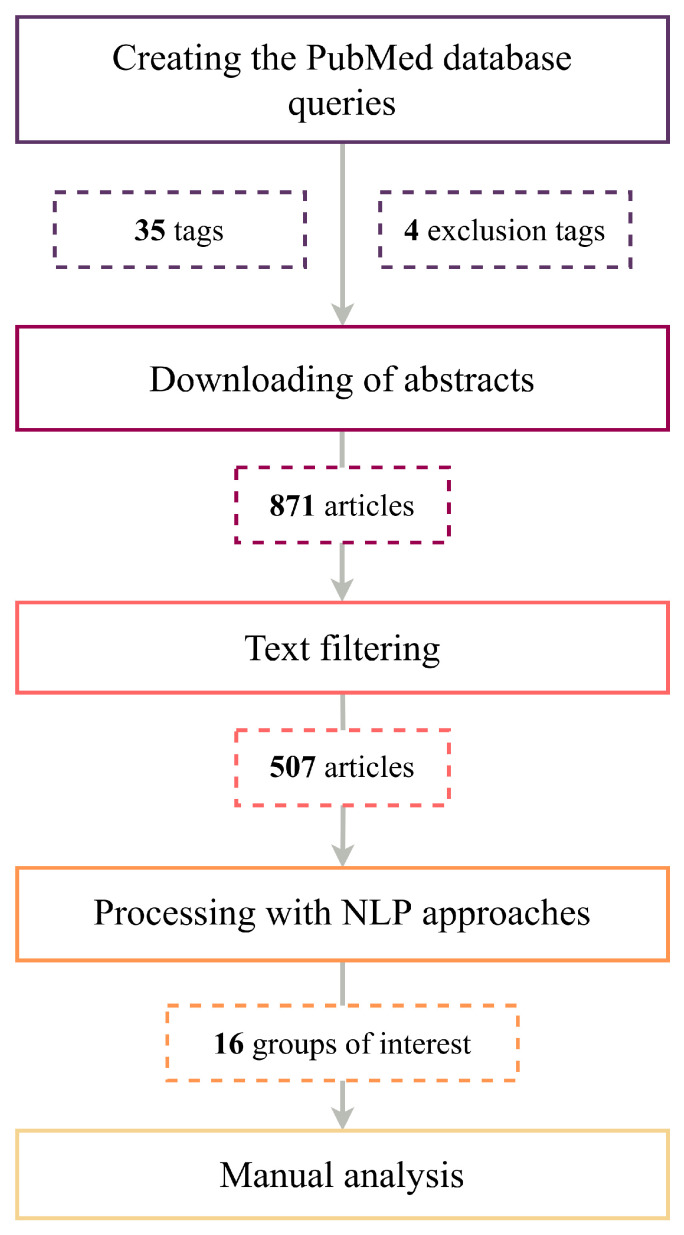
The stages of the pipeline for article processing for the literature review.

**Figure 2 metabolites-14-00054-f002:**
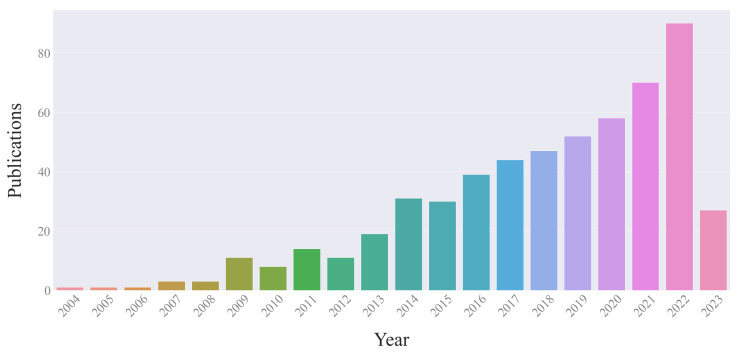
Number of publications in various metabolomics areas, including studies on lipidomics, glycomics, and amino acids per year in 2004–2023.

**Figure 3 metabolites-14-00054-f003:**
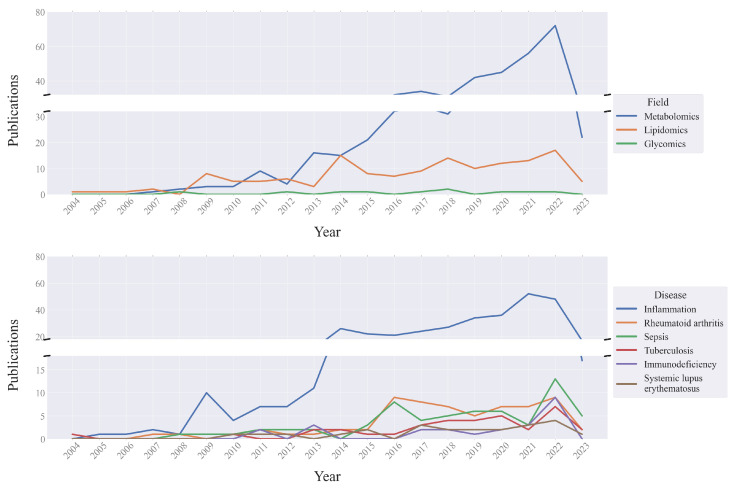
Number of publications per year for each field (**top picture**) and disease (**bottom picture**).

**Table 1 metabolites-14-00054-t001:** A list of keywords for a literature search. The list is divided into topics for better comprehension.

Fields	Field Tags	Diseases	Biomaterial	Methodology	Exception Tags
Metabolomics	metabolomics, metabonomics, metabolic profiling, metabolomics profile	Systemic lupus erythematosus Sepsis Rheumatoid arthritis Tuberculosis Immunodeficiency Immunodeficient state Inflammation	blood serum plasma	liquid chromatography gas chromatography mass spectrometry LC-MS HPLC-MS HPLC-MS/MS UPLC-MS UHPLC-MS GC-MS	genomics genetics food markers pharmacology
Lipidomics	lipidomics, lipidome, lipid profiling				
Glycomics	glycomics, glycome				

**Table 2 metabolites-14-00054-t002:** Number of papers on the most socially important non-cancer diseases in metabolomics, lipidomics, and glycomics fields from the last 10 years.

	Metabolomics	Lipidomics	Glycomics	Total
Inflammation	223	91	4	318
Rheumatoid arthritis (RA)	48	9	2	59
Sepsis	45	9	1	55
Tuberculosis	32	0	1	33
Immunodeficiency	19	3	0	22
Systemic lupus erythematosus (SLE)	19	1	0	20
Total	386	113	8	

## Data Availability

No new data were created in this review.

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
