# Peer review of "Advances in Mass Spectrometry-Based Blood Metabolomics Profiling for Non-Cancer Diseases: A Comprehensive Review"

_metabolites, 2024, doi:10.3390/metabo14010054_

Round 1

Reviewer 1 Report

Comments and Suggestions for Authors

The authors comprehensively summarized the recent developments in the metabolite analysis from blood. The metabolite related to the various cancer types was previously studied. However, the non-cancer metabolites are still poorly explored. In the current review, the authors provided metabolite technology identification approaches with their recent developments. The authors also described the multiple disease-related metabolite identification studies which are helpful for the researchers.

This review has to go through extensive revision before publication. I have listed a few major points that could help to improve it further.

1.     In the beginning, the authors did a thorough literature search using specific keywords in a specific time period. This could potentially deviate from the main focus of this review is on metabolites identified in various diseases. As the metabolite field is rapidly developing and the increasing trend is quite expected. The authors could remove this section or transfer it to supporting information.

2.     Mass spectrometry and chromatography should be two different sections outlining the  separation methods and their developments, ionization methods, detection methods and their respective data analysis. Capillary electrophoresis-mass spectrometry (CE-MS) is an entirely different method of separation.

3.     The MS data processing should outline the vendor-specific data processing pipeline. Every mass spec vendor has specific file formats and preprocessing steps.

4.     In the “ Research field landscape” the authors are talking about lipidomics and glycomics which could be potentially important and interesting. However, these fields are vast and need more detailed review on them. The authors could make a single section of them or remove it.  “Other metabolite” section should be included as the main section.

5.     In the “Disease Study Landscape”, the authors comprehensively listed multiple diseases including metabolic and infectious diseases. The metabolite profile of each disease is different and certain biomarkers are exclusively expressed in certain health conditions. The authors may want to separate infectious diseases and metabolic diseases with additional information and add more diseases.

6.     The authors may want to add future perspectives in the conclusion section as this is a rapidly growing field and the trend in its development needs to be envisioned to provide guidance to the field.

Reviewer 2 Report

Comments and Suggestions for Authors

In the presented script Demicheva et al. review the emerging role of mass spectrometry-based metabolomics in non-cancer diseases. The authors clearly describe the criteria that led to the inclusion of papers into their review. Indeed, the combination of searches with LLM helps to create an unbiased and broad selection of manuscripts, which the authors condense in a well written and highly informative review. A few points could be added to complete the manuscript:

11)      The authors primarily describe triple quad and QTOF instruments. Other high resolution mass spectrometers such as orbitraps should be included in the introduction.

22)      APCI sources could be included.

33)      The authors include SRM and MRM techniques, but more modern PRM should be mentioned, too.

Reviewer 3 Report

Comments and Suggestions for Authors

Ekaterina Demicheva et al review the advances in mass spectrometry-based blood metabolomics profiling for non-Cancer diseases. Overall, this manuscript is too long and covered a wide range of topics. The authors are suggested to shorten it and my main comments are as bellow.

1.     Figure/table legends are suggested so that the figures/tables can be understood without looking into the main manuscript.

2.     The authors are suggested to discuss and highlight the roles of metabolomics in non-cancer disease studies, such as biomarkers of prognosis and diagnosis, disease progression, therapeutic mechanism et al, as well as the advantage and limitation of metabolomics compared to other omics approaches.

Comments on the Quality of English Language

The manuscript is suggested to be proof-read by native English speakers.

Round 2

Reviewer 1 Report

Comments and Suggestions for Authors

The authors revised the manuscript thoroughly and now looks much improved. A few more minor changes are needed.

1. Separation methods should come first followed by Mass spectrometry methods.

2.  The authors listed the extensive history of all diseases listed. It could be shortened to a brief background, disease mechanism, and specific biomarker discovery. 

3. Change section "7. Conclusion" to 7. Conclusion and Future Perspectives and reorganize the paragraphs.

Comments on the Quality of English Language

Need to check the grammar and spelling. 

Reviewer 3 Report

Comments and Suggestions for Authors

The review revised the manuscript carefully and addressed most of the questions, I would suggest the manuscript is ready to be published.
